# STANCE: a unified statistical model to detect cell-type-specific spatially variable genes in spatial transcriptomics

Haohao Su [1], Yuesong Wu[1], Bin Chen [2,3,4] & Yuehua Cui [1] ✉

One of the major challenges in spatial transcriptomics is to detect spatially variable genes (SVGs), whose expression patterns are non-random across tissue locations. Many SVGs correlate with cell type compositions, introducing the concept of cell type-specific SVGs (ctSVGs). Existing ctSVG detection methods treat cell type-specific spatial effects as fixed effects, leading to tissue spatial rotation-dependent results. Moreover, SVGs may exhibit random spatial patterns within cell types, meaning an SVG is not always a ctSVG, and vice versa, further complicating detection. We propose STANCE, a unified statistical model for both SVGs and ctSVGs detection under a linear mixed-effect model framework that integrates gene expression, spatial location, and cell type composition information. STANCE ensures tissue rotation-invariant results, with a two-stage approach: initial SVG/ctSVG detection followed by ctSVG-specific testing. We demonstrate its performance through extensive simulations and analyses of public datasets. Downstream analyses reveal STANCE's potential in spatial transcriptomics analysis.

Spatial transcriptomics (ST) has garnered rapidly increasing attention in biology. Technologies such as 10x Visium[1], Slide-Seq[2], and MERFISH[3] enable the profiling of the transcriptome at various spatial resolutions, facilitating the investigation of spatial expression variation, cell-cell communication, and many other areas of genome biology.

One significant challenge in analyzing spatial transcriptomics data is to effectively and efficiently detect spatially variable genes (SVGs), whose gene expression displays non-random spatial patterns in tissue sections. The characteristics of SVGs indicate their critical roles in deciphering spatial variations in gene expression across a tissue section. Many computational methods have been developed to address this challenge. Existing statistical methods for detecting SVGs can be classified into two groups: (1) spatial regression methods, which treat spatial coordinates as fixed effects with various transformations or basis expansions[4–6], and (2) distance-based methods, which use a spatial kernel to convert spatial dissimilarity into a similarity matrix, then estimate the spatial variance component for further testing[7]. Methods such as SpatialDE[8], SPARK[9], and nnSVG[10] fall into this

category. SPARK-X[11] follows the same principle, but assesses how the similarity between spatial locations corresponds to gene expression similarity based on a nonparametric test. Readers are referred to Yan et al. (2024)[7] for a comprehensive review of 31 methods for SVG detection.

In both sequencing- and imaging-based ST data, gene expression variation across spatial spots often arises from differences in expression levels among distinct cell types. Since cell types are unevenly distributed within tissues, disregarding cell type information can obscure spatial patterns that extend beyond those driven by cell-type composition[7]. Moreover, many SVGs with spatial expression variation are closely associated with cell type categories or compositions, giving rise to the concept of cell-type-specific spatially variable genes (ctSVGs). ctSVGs are characterized by non-random spatial expression patterns within specific cell types. Notably, an SVG may exhibit random spatial patterns within individual cell types, and conversely, a ctSVG may appear spatially random when considered more broadly, emphasizing that an SVG and a ctSVG are not inherently

[1]Department of Statistics and Probability, Michigan State University, East Lansing 48824 MI, USA. [2]Department of Pharmacology and Toxicology, Michigan State University, East Lansing 48824 MI, USA. [3]Department of Computer Science and Engineering, Michigan State University, East Lansing 48824 MI, USA. [4]Department of Pediatrics and Human Development, Michigan State University, Grand Rapids 49503 MI, USA. ✉e-mail: cuiy@msu.edu

interchangeable[4]. This distinction highlights the necessity of specialized methods for ctSVG detection, which rely on integrating ST data with external cell-type annotations for spatial spots.

Currently, there are three statistical methods available for detecting ctSVGs[7], all of which use cell type-specific fixed effects to model spatial location information and apply a regression model to link these fixed effects to gene expression patterns. Statistical hypothesis tests are then conducted based on these fixed effects to detect ctSVGs. Here, we briefly introduce these three methods.

CTSV[4] is a statistical tool designed to detect ctSVGs using row count data from spatial transcriptomics. Leveraging cell-type composition information from deconvolution, CTSV fits a zero-inflated negative binomial regression model and treats the two spatial coordinates as two separate covariates, denoted as $h_1$ and $h_2$, to assess spatial effects. To identify ctSVGs for a specific cell type $k$, statistical inference is performed on the coefficients $\beta_{k1}$ and $\beta_{k2}$, which correspond to the fixed effects for $h_1$ and $h_2$, respectively, by testing whether they are zero for each gene. A gene is considered cell type-specific if at least one of the effects is statistically significant. The authors also considered the exponential transformation of the spatial coordinates as new covariates.

C-SIDE[5] is another statistical method used to identify cell type-specific differential expression in spatial transcriptomics. C-SIDE employs a Poisson regression model to capture the spatial pattern of gene expression by introducing basis expansion of spatial coordinates. The authors introduce $L$ smooth basis functions, whose linear combinations represent the overall smooth gene expression functions across spatial locations. Each basis function is tested using a two-sided z-test, and the $L$ p-values are then adjusted and the minimum p-value is chosen as the final one to declare significance. However, selecting the appropriate number of basis functions $L$ can be challenging due to the lack of a consistent rule. The default number is set as $L = 15$ which may be too large, leading to potentially overfitting with a large number of degrees of freedom for testing, and consequently low statistical testing power.

spVC[6] also applies a Poisson regression model to capture the spatial pattern of gene expression but constructs fixed effects differently. spVC uses a two-stage procedure to detect ctSVGs. In the first stage, $K$ cell type-specific non-spatial effects corresponding to $K$ cell types, along with a residual spatial effect, are introduced into the model and tested to determine if they are equal to zero. Genes for which the residual spatial effect and at least one cell type-specific non-spatial effect are not zero are considered SVGs, indicating that at least one cell type has a constant effect on gene expression. In the second stage, a full model is fitted for these genes by adding another $K$ cell type-specific spatial effects, which are approximated through the bivariate penalized spline approximation. For each gene and each cell type of interest, the associated cell type-specific spatial effect is tested to determine if the gene is an SVG specific to that cell type. However, as noted in CTSV, an SVG does not necessarily have to be a ctSVG, and vice versa. A gene can be a ctSVG but not an SVG when the expressions of different cell types are mixed together. The first stage of spVC only checks whether the target gene is an SVG. Therefore, only those ctSVGs that are also SVGs would be identified by spVC.

The aforementioned three statistical methods for ctSVGs detection treat cell type-specific spatial effects as fixed effects during modeling. Thus, they all suffer from a critical issue: the hypothesis testing results are not invariant to the rotation of spatial coordinates. This means that if the tissue samples were examined at a different angle from the original one, the testing results could be very different. Tissue sections are often positioned randomly during sample preparation (as an example, see Supplementary File Fig. S1 for four MOB sample replicates from the same tissue), and different orientations give different spatial coordinates which fail methods that treat spatial coordinates as fixed effects, and consequently leading to high false positives or false negatives. This issue is particularly problematic in the context of multi-slice spatial transcriptomics data analysis, where slices are not spatially aligned. Ensuring rotation invariance is critical for robust multi-slice integration. Consequently, treating spatial coordinates (and their transformations) as fixed effects is statistically unsound, highlighting the need for methods that account for rotation invariance when modeling cell type-specific spatial effects.

Additionally, the complicated relationship between overall SVGs and ctSVGs also poses challenges in real analysis when dealing with SVG or ctSVG detection. In this work, we proposed STANCE (Spatial Transcriptomics ANalysis of genes with Cell-type-specific Expression), a unified statistical method that can identify both SVG and ctSVG. In the first stage, STANCE performs an overall test to identify the presence of SVGs and ctSVGs, classifying genes that pass this test as unified-type SVGs (utSVGs), which encompass both SVGs and ctSVGs. These utSVGs are then subjected to a second stage analysis for ctSVG detection. Our method uses spatial kernel matrices that rely solely on the relative distance between spatial spots and guarantees that the testing results are invariant to spatial rotation and transformation. Through extensive simulation studies and analysis of three real datasets, we demonstrate the utility of our method which greatly enriches the toolbox of ST data analysis.

## Results
### Method overview and simulations

The schematic overview of STANCE is displayed in Fig. 1. The technical details are provided in the Methods section and Supplementary File. We developed a two-stage testing procedure for STANCE: (1) Stage 1 test: the STANCE overall test for utSVGs and (2) Stage 2 test: the STANCE individual test for ctSVGs. Rejecting the overall test leads to the conclusion of utSVGs which include both SVGs and ctSVGs. We then proceed to test if a utSVG is a ctSVG by the stage 2 test. Downstream analyses with STANCE include generating variance plots to display the relative variance of a ctSVG across different cell types, performing spatial domain detection using utSVGs or ctSVGs, and conducting functional enrichment analysis to gain further biological insights.

We conducted a simulation study to evaluate the statistical robustness of existing methods, C-SIDE, spVC, and CTSV, in the detection of ctSVG (Simulation details are provided in Supplementary File "Spatial Rotation Simulation" section). These methods treat spatial locations as fixed effects, which compromises spatial rotation-invariance in their analyses. Our findings revealed inconsistencies in testing results for these methods before and after spatial rotations at various angles, despite the expectation that outcomes should remain unaffected by such rotations. Notably, testing results varied systematically with changes in the rotation angle. These inconsistencies underscore the unreliability of these methods for ctSVG detection, indicating their unsuitability for robust ctSVG analysis.

We also designed two sets of simulations to assess the efficacy of STANCE and compared it to existing methods in detecting spatial variability in spatial transcriptomics data (Simulation details are provided in METHODS). Figure 2 displays the scenarios of different cell type compositions and the expression pattern images of different domains from different cell types of a representative gene. As STANCE relies on cell type composition information for detecting SVGs and ctSVGs, we tested STANCE using both true cell type composition (STANCE-oracle) and deconvolved cell type composition from the reference-based deconvolution method RCTD[12] (STANCE-RCTD). We also evaluated the false positive control of STANCE when the cell type composition information is misspecified.

The first set of simulations evaluated the performance of the STANCE overall test for utSVG detection. We compared it with SPARK-G (the Gaussian version of SPARK) and SPARK-X, both of which are designed to detect SVGs.

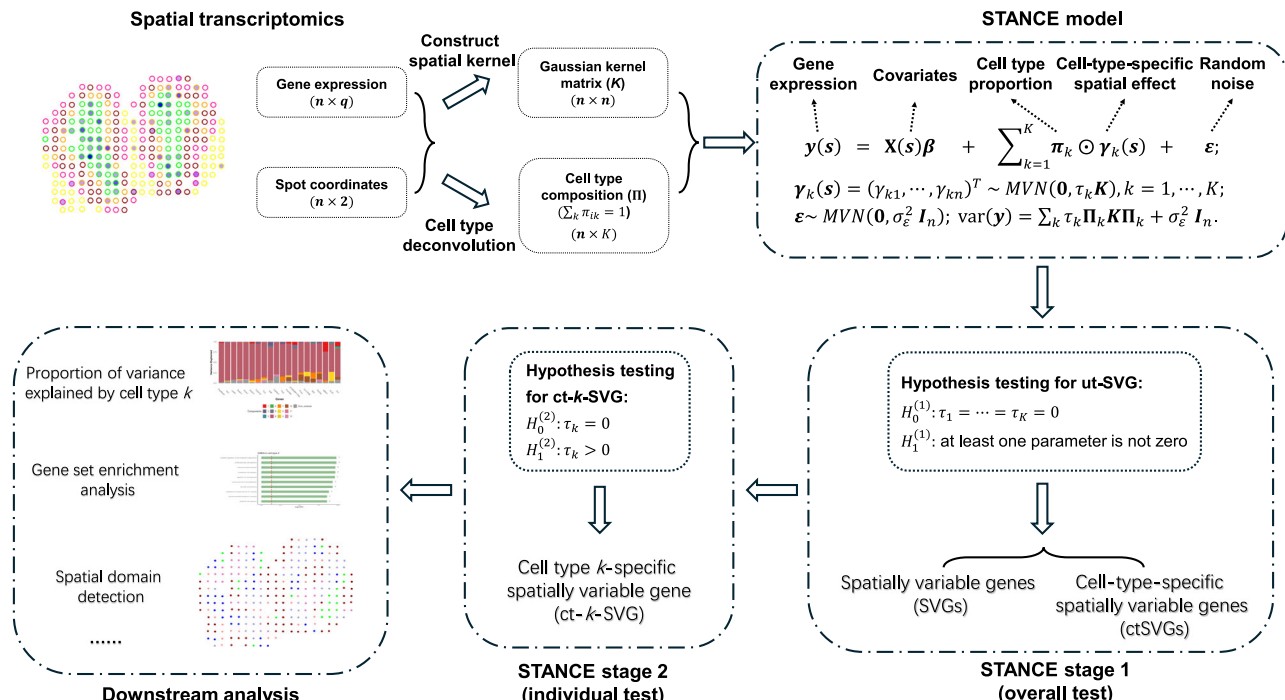

**Fig. 1 | Overview of STANCE.** STANCE is a unified statistical model designed to detect both spatially variable genes (SVGs) and cell-type specific spatially variable genes (ctSVGs) in spatial transcriptomics. Utilizing cell type composition information obtained from cell type deconvolution, STANCE constructs a kernel matrix that links gene expression patterns with spatial locations through a linear mixed effect model incorporating $K$ cell type-specific spatial random effects. In the 1st step, both SVGs and ctSVGs are identified through an overall test, which determines if all $K$ cell-type specific spatial variance components are non-zero. Genes passing this unified test are termed utSVGs. In the 2nd step, for each cell type of interest, ctSVGs are determined from the utSVGs by conducting individual tests to check if the corresponding spatial variance component is non-zero. The relative variance contribution of cell types can be visualized through a stacked variance plot. Downstream analysis of STANCE includes spatial domain detection using utSVG or ctSVG and cell-type specific gene set enrichment analysis.

Under the null of no spatial genes (Supplementary File Fig. S2), STANCE-oracle effectively controlled the type I error across all the scenarios and under different data dispersions (Fig. 3a). The figures also show the empirical type I error rates for different methods. When using the deconvolved cell type compositions with RCTD, STANCE-RCTD produced testing results comparable to STANCE-oracle, indicating STANCE's robustness through deconvolution. In each scenario, SPARK-G also performed well, though its results were slightly conservative. Conversely, the p-values produced by SPARK-X were much more conservative, potentially resulting in low statistical testing power. SPARK-X gains computational efficiency by suffering from power loss, a result aligning well with the original observation of SPARK-X. Overall, there was no significant difference in the testing results under different scenarios of cell type compositions.

We also conducted a sensitivity analysis for type I error rate control under misspecified cell type compositions. Instead of deconvolving to three cell types (the grand truth), we assume there were four cell types with equal proportions across the spatial locations. Under the misspecified cell type compositions, STANCE-misspecified can still effectively control the type I error across different settings (Supplementary File Fig. S3), showcasing the robustness of STANCE in type I error control. Additionally, there was no significant difference in testing results in different scenarios.

Alternative Case 1 considered genes that were both SVGs and ctSVGs, i.e., utSVG (Supplementary File Fig. S4). Under this case, STANCE-oracle, STANCE-RCTD, and SPARK-G exhibited comparable testing power across a range of FDR values, while SPARK-X was less powerful in detecting utSVGs (Fig. 3b). The fold change in expression mean significantly impacted testing power, with power increasing as the relative distance of gene expression mean from the baseline increased (Supplementary File Fig. S5). Additionally, power increases as the dispersion increases (larger dispersion means lower variance).

In Alternative Case 2, which involved only ctSVGs (Supplementary File Fig. S6), SPARK-X nearly had no testing power, and SPARK-G performed slightly better (Fig. 3c). This is expected as both were not designed to detect ctSVGs. STANCE-oracle outperformed STANCE-RCTD under low dispersion, indicating that the accuracy of deconvolution has an impact on the statistical power of STANCE. But when the dispersion increased to 1.5, deconvolution had little impact on the testing power. Additionally, STANCE became more powerful with an increased proportion of the cell type corresponding to the ctSVGs (Scenario 3).

In Alternative Case 3, involving only SVGs (Supplementary File Fig. S7), both STANCE-oracle and STANCE-RCTD demonstrated superior performance compared to SPARK-G and SPARK-X (Fig. 3d), even though detecting SVGs is the primary function of SPARK-G and SPARK-X. STANCE-oracle and STANCE-RCTD yielded closely matched testing results, indicating its robust performance for SVG detection. Similar to Alternative Cases 1 and 2, the statistical power of detecting SVGs increased with increased dispersion. Additionally, the mean fold change from the baseline expression significantly affected testing power; as the fold change deviated away from 1, the statistical power also increased (Supplementary File Fig. S8). The results show that STANCE is more effective in scenarios with higher fold changes in expression levels and lower variance.

The power analysis results across all three alternative cases demonstrate a common trend: statistical power increases as the variance decreases (or the dispersion increases). This observation is biologically and statistically intuitive. Lower variance in gene expression reduces noise, making the differences in expression levels between distinct spatial domains more consistent and reliable. As a result, the underlying gene expression patterns become more pronounced and easier to detect. This enhanced stability allows for more robust identification of differentially expressed genes associated with specific

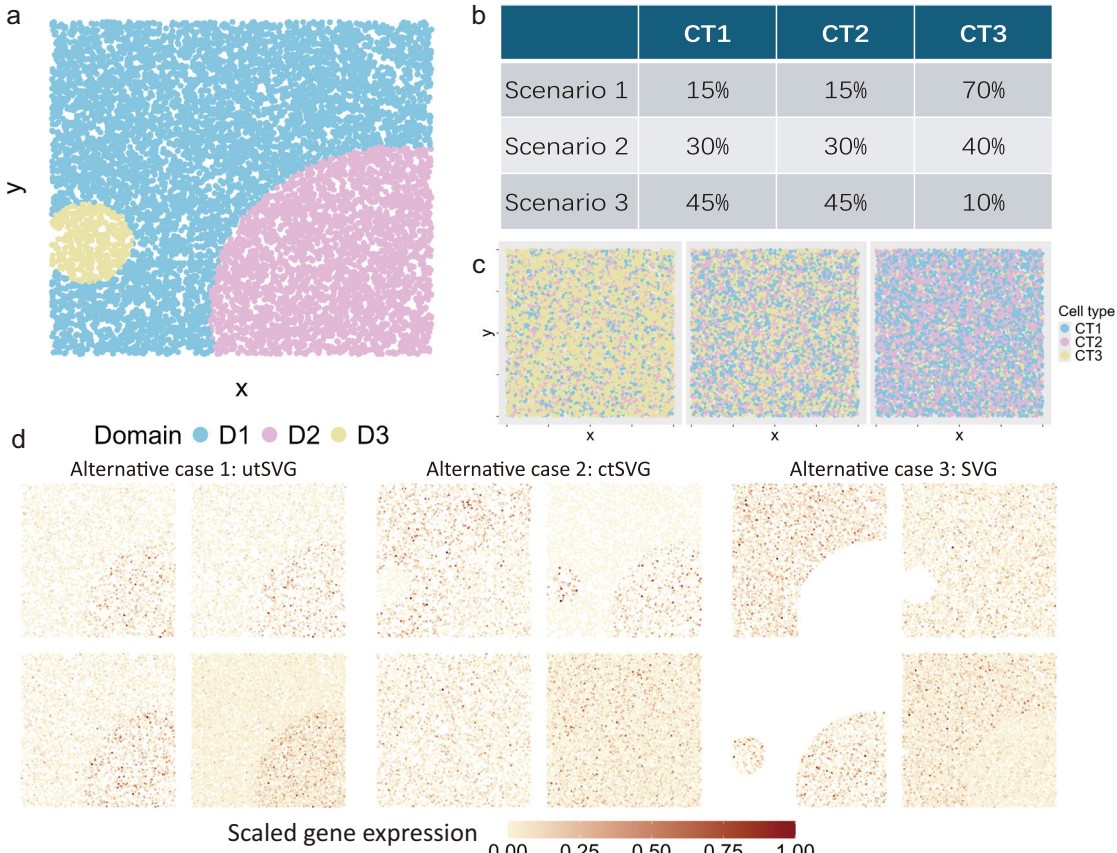

**Fig. 2 | Overview of Simulation 1 and schematic illustration of different cases of ctSVG and SVG. a** The simulated tissue contains three domains (D1, D2 and D3) whose location and size are randomly generated (light blue = D1, light purple = D2, and light yellow = D3). **b** Spatial pattern of random cell type distributions under the three scenarios. Left: Scenario 1. Middle: Scenario 2. Right: Scenario 3. **c** The three scenarios of cell type composition. **d** The spatial expression pattern of a representative gene. For each case, the four patterns correspond to Cell Type 1-specific pattern (top left); Cell Type 2-specific pattern (top right); Cell Type 3-specific pattern (bottom left); and the combined pattern (bottom right). Alternative case 1 shows that a gene is both SVG and ctSVG. Alternative case 2 shows that a gene is cell-type 1 and 2 ctSVG but not cell-type 3 ctSVG and not SVG. Alternative case 3 shows that a gene is not ctSVG but is SVG.

spatial regions. In biological systems, reduced variability often reflects tightly regulated processes or well-defined spatial organization, which can further amplify the signal-to-noise ratio and improve the detectability of spatially variable genes. Thus, the observed relationship between dispersion, variance, and power aligns with expectations from both biological mechanisms and statistical principles.

The second set of simulations aimed to evaluate the performance of the STANCE individual test in detecting ctSVGs. Although the ctSVG detection method spVC is not spatial rotation-invariant and not suitable for ctSVG detection, we still include it into the comparison. Figure 4 displays the composition of different ctSVGs and the expression pattern images of a representative gene under different cases.

Essentially, STANCE-oracle effectively controlled the type I error across two different dispersion settings (Fig. 5a). When using the deconvolved cell type compositions through RCTD, STANCE-RCTD produced testing results comparable to STANCE-oracle that used true cell type compositions, indicating STANCE's robustness under deconvolution. In contrast, both spVC-oracle and spVC-RCTD exhibited slightly inflated p-values under true null scenarios, indicating weaker type I error control compared to STANCE. These findings highlight STANCE's superior ability to maintain statistical rigor in type I error control. Additionally, the analysis revealed no significant differences in type I error control across the different dispersion settings, further showcasing the robustness of both STANCE configurations to variations in data dispersion.

When detecting ctSVGs, STANCE-oracle exhibited strong testing power across a wide range of FDR levels. Although the power of STANCE-RCTD was slightly lower than that of STANCE-oracle due to the reduced accuracy of deconvolved cell type compositions, it still performed robustly. In comparison, STANCE consistently outperformed spVC in detecting ctSVGs across nearly all scenarios of data dispersion and cell type proportions. Notably, the power to detect ctSVGs increased as the proportion of the cell types corresponding to the ctSVGs increased (Fig. 5b). This trend is reasonable, as higher proportions provide more information, thereby enhancing the detection capability. Additionally, we observed that statistical power improved with higher dispersion settings (low variances), further underscoring the effectiveness of STANCE in various data conditions. These results collectively highlight the advantages of STANCE in detecting ctSVGs, particularly under diverse and challenging conditions.

### Real data analysis: human breast cancer data
We applied STANCE to three ST datasets to demonstrate its performance and compared the results with other methods such as SPARK-G and SPARK-X. Additional details about the data sources and data processing can be found in Supplementary File.

We first examined the 10x Visium spatial transcriptomics dataset from the human HER2-positive breast cancer tumors[13]. We used sample H1 as the main analysis example, which contains gene expression count data for 15,030 genes and 613 spots. The original study applied Stereoscope[14], a scRNA-seq reference-based deconvolution method, to

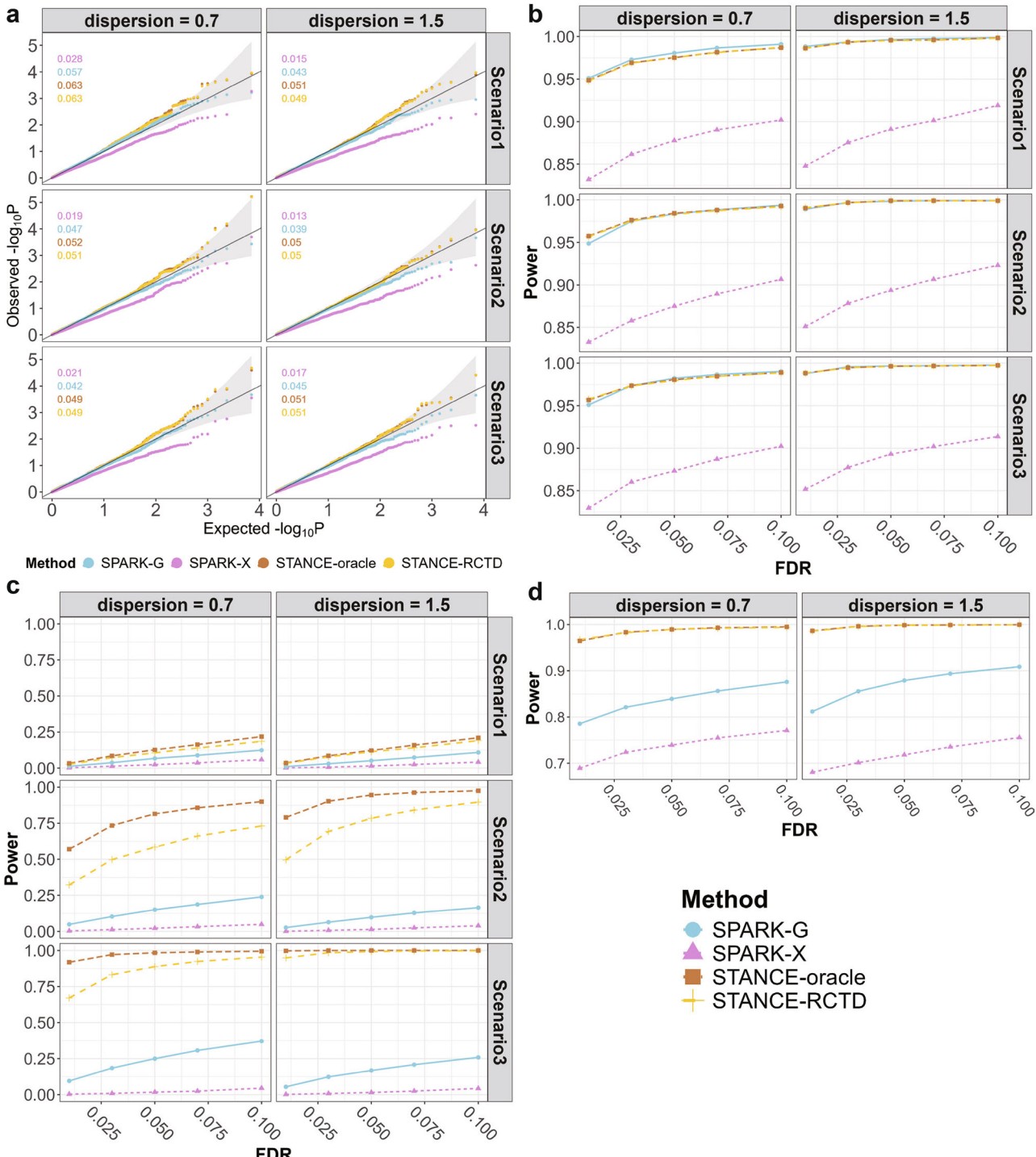

**Fig. 3 | Results of Simulation 1. a.** Simulation results under Simulation 1 null case with the four methods. The Q-Q plots of the observed $-\log_{10}p$ against the expected $-\log_{10}p$ for different methods are displayed across various dispersion parameters and scenarios, where $p$ represents the p-value from the score test. False positive rates (type I error rates) for each method are displayed in the top-left corner of each panel, with colors corresponding to the legend. The light gray region represents the 95% error band, illustrating the range in which 95% of points are expected to fall under the null hypothesis, assuming the p-values follow the expected uniform distribution. **b** Simulation results under Simulation 1 alternative case 1, which involves both SVGs and ctSVGs. The plots display power values across a range of false discovery rates for different methods, considering various dispersion parameters and cell-type composition scenarios. **c** Simulation results under Simulation 1 alternative case 2, which involves only ctSVGs. The plots display power values across a range of false discovery rates for different methods, considering various dispersion parameters and cell-type composition scenarios. **d** Simulation results under Simulation 1 alternative case 3, which involves only SVGs. The plots display power values across a range of false discovery rates for different methods, considering various dispersion parameters. Source data are provided as a Source Data file.

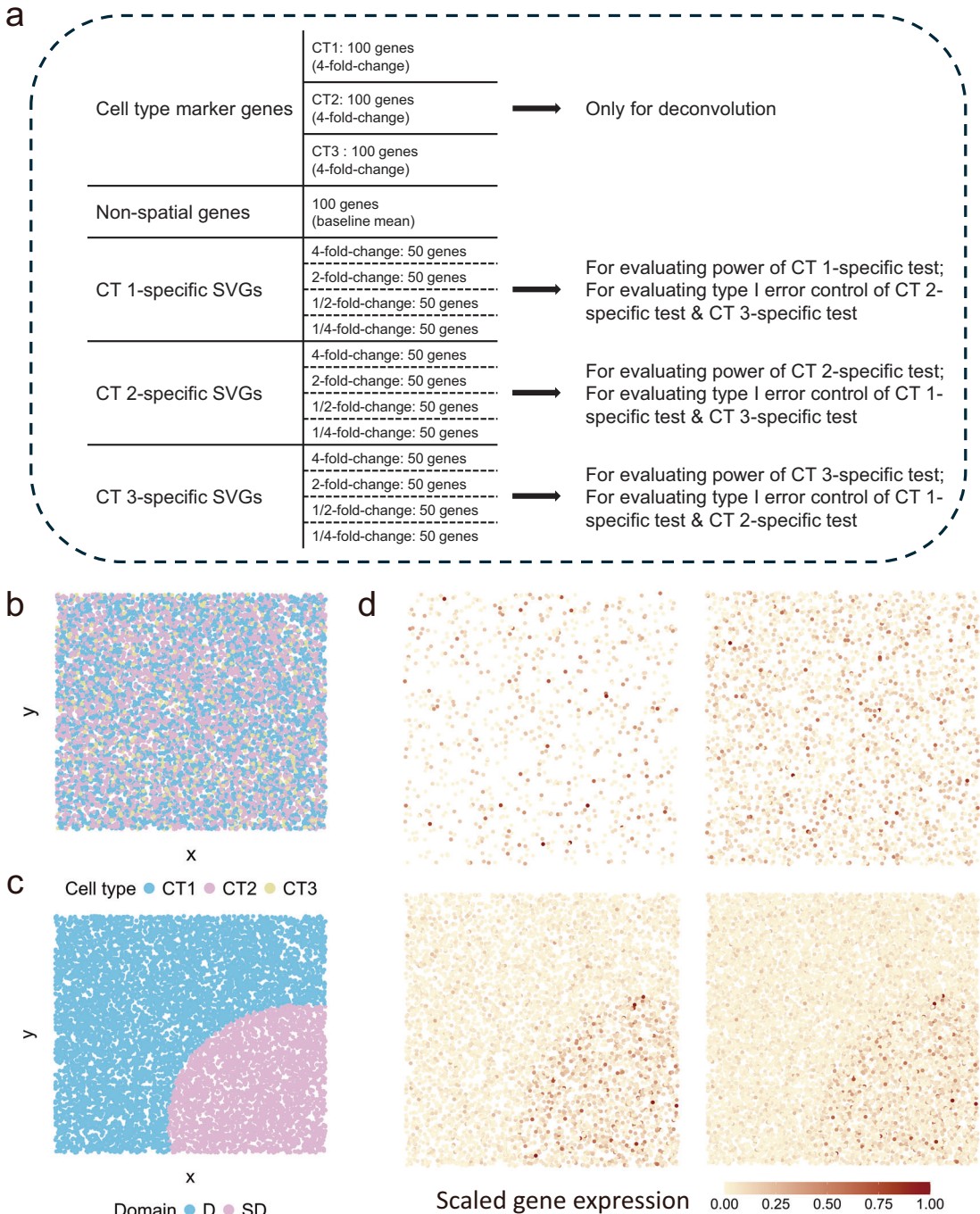

**Fig. 4 | Overview of Simulation 2. a** The scheme of Simulation 2. Under each dispersion parameter, 1,000 genes are generated, where 300 of them are cell type marker genes and 600 of them are ctSVGs. Specifically, each cell type has 100 distinct marker genes and 200 ctSVGs. For each cell type, the testing results of its associated individual test on its ctSVGs are used to evaluate type I error, while the testing results of the other two cell type-specific individual tests on its ctSVGs are used to evaluate testing power. **b** Spatial pattern of cell type composition. **c** The simulated tissue contains two domains whose location and size are randomly generated. **d** The spatial patterns of single-cell gene expression of a representative gene. Top left: cell type 1-specific pattern (not a ctSVG). Top right: cell type 2-specific pattern (not a ctSVG). Bottom left: cell type 3-specific pattern (ctSVG), high expression in domain *SD* than in domain *D*. Bottom right: the combined pattern (SVG), high expression in domain *SD* than in domain *D*.

annotate the cell type composition of each spot. We used the major tier annotation with 8 cell types, including myeloid cells, T cells, B cells, epithelial cells, plasma cells, endothelial cells, cancer-associated fibroblasts (CAFs), and perivascular-like cells (PVL cells). We then removed 21 genes identified as ring-pattern technical artifacts in the original study following[14], low-expressed genes that do not express at more than 10% spots, resulting in 10,053 genes.

This dataset contains spatial domain annotation by pathologists based on H&E images, consisting of in situ cancer, invasive cancer, breast glands, adipose tissue, immune infiltrate, connective tissue, and other spots in the undetermined region. After removing spots with less than 10 total counts and genes expressed in less than 10% spots following the SPARK's QC steps, we ended up with 607 spots and 2816 genes for further analysis.

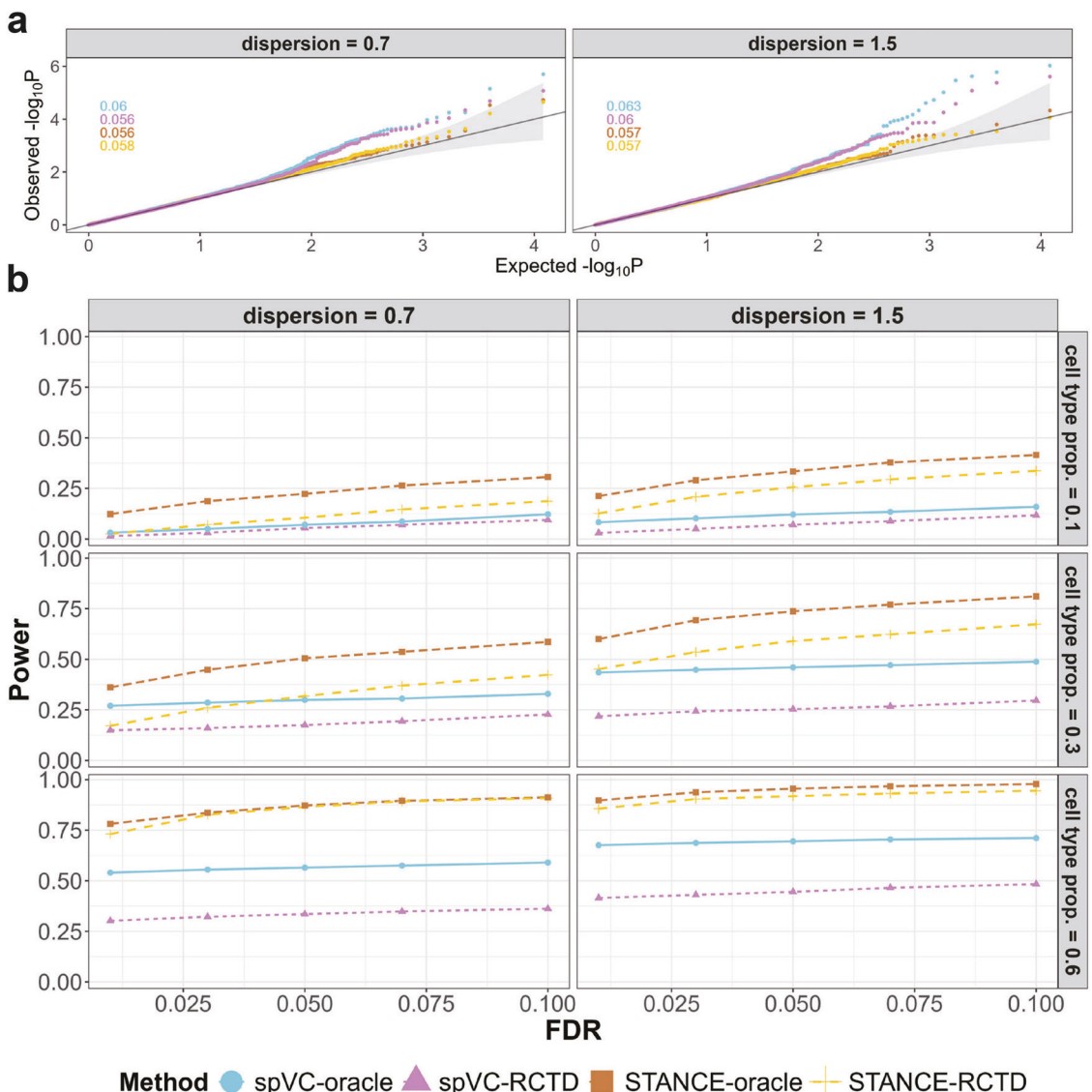

**Fig. 5 | Results of Simulation 2. a** Simulation results of type I error control under Simulation 2. The Q-Q plots of the observed $-\log_{10}p$ against the expected $-\log_{10}p$ for different methods are displayed across various dispersion parameters, where $p$ represents the $p$-value from the score test. False positive rates (type I error rates) for each method are displayed in the top-left corner of each panel, with colors corresponding to the legend. The light gray region represents the 95% error band, illustrating the range in which 95% of points are expected to fall under the null hypothesis, assuming the p-values follow the expected uniform distribution. **b** Simulation results of testing power under Simulation 2. The plots display testing power across a range of FDR levels for different methods, considering various dispersion parameters and cell type proportions. Source data are provided as a Source Data file.

We applied STANCE, SPARK-G and SPARK-X to this dataset. The STANCE overall test identified 330 genes, with p-values adjusted by the Benjamini-Yekutieli method[15] to meet an FDR of $p = 0.05$. These genes are considered utSVGs, a mixture of SVGs and ctSVGs. SPARK-G identified 315 SVGs, of which 254 were also identified by the STANCE overall test. SPARK-X identified 1470 SVGs, with 299 overlapping with those detected by STANCE (Supplementary File Fig. S11). For the 330 utSVGs detected by the STANCE overall test, we further conducted cell-type-specific individual tests for each of the 8 cell types, and identified a total of 286 ctSVGs across all 8 cell types. The number of ctSVGs in each cell type is shown in Fig. 6a. Epithelial cells has the most ctSVGs with 204 genes, while endothelial cells has the fewest, with 19 genes. Some of the ctSVGs show significant expression in different cell types and overlapping results are shown in Fig. 6b. Among which, cell type CAFs had the least overlapped genes with other cell types. This may be due to the specialized role of CAFs in the tumor microenvironment. CAFs are involved in extracellular matrix remodeling, immune modulation, and supporting cancer progression, leading to the expression of unique genes tailored to these functions. This distinct gene expression pattern reflects the specific signaling pathways and interactions CAFs engage in within their niche, making their gene profile less similar to other cell types. spVC was also applied to this dataset as a comparative method. Interestingly, it failed to identify any ctSVGs, as no genes passed its initial filtering step. This unexpected outcome suggests that spVC may exhibit excessive conservativeness in certain datasets or analysis scenarios, particularly during its initial screening process. Such stringency could limit its ability to detect ctSVGs under specific conditions, potentially overlooking biologically meaningful results.

We further analyzed the 286 ctSVGs detected by STANCE. The spatial domain analysis illustrated that important biological signals, absent in SVGs, are well-preserved in utSVGs, the combination of SVGs and ctSVGs. We compared domain detection accuracy using utSVGs and ctSVGs identified by STANCE, as well as SVGs detected by SPARK-

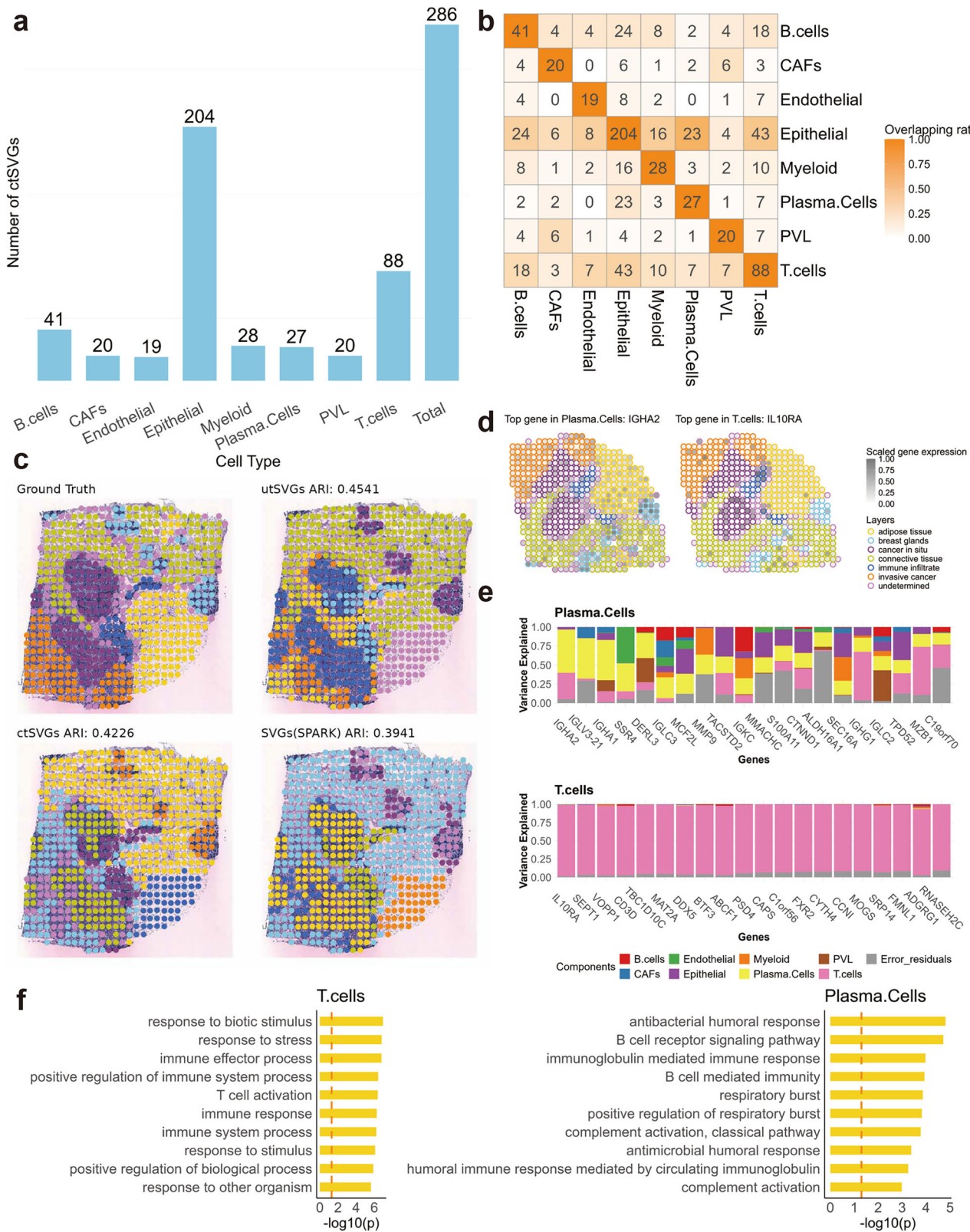

G, as input features. SpatialPCA[16], SeuratPCA[17], BayesSpace[18], stLearn[19], and SpaceFlow[20] were applied as domain detection algorithms. The adjusted Rand index (ARI)[21] was used for evaluation, following the SpatialPCA's procedure, excluded the undetermined region and treated the remaining regional annotations as ground truth. Across all conditions, utSVGs and ctSVGs consistently outperformed SPARK-G SVGs. We present the SpatialPCA results (Fig. 6c) as an illustration due

to its superior domain detection performance, with results from other algorithms provided in Supplementary File Fig. S12.

Using SpatialPCA, we achieved an ARI of 0.45 with utSVGs (comprising both SVGs and ctSVGs) identified by STANCE. When ctSVGs alone were used as input features, the ARI was 0.42, while SPARK-G SVGs yielded the lowest accuracy with an ARI of 0.39. SPARK-G SVGs failed to distinguish between connective tissue and invasive cancer in

**Fig. 6 | Results of the Her2+ breast cancer data. a** The bar plot displays the numbers of detected ctSVGs for each cell type. **b** The heatmap displaying the intersections of ctSVGs between cell types. **c** Domain detection results display the true and estimated domain annotations by SpatialPCA. Top left: Ground truth. Top right: Based on utSVGs by STANCE. Bottom left: Based on ctSVGs by STANCE. Bottom right: Based on SVGs by SPARK-G. **d** Spatial pattern plots of scaled gene expression for two ctSVGs (Left: *IGHA2*; Right: *IL10RA*). Spots are outlined with colors indicating the domain annotation. **e** The stacked bar plots display the

variance explained by 8 cell type-specific spatial effects and random error, illustrating the top 20 ctSVGs for each cell type. Top: Plasma cells. Bottom: T cells. **f** The gene set enrichment analysis results on two groups of ctSVGs detected by STANCE. The top 10 significant pathways are shown. The enrichment is given as $-\log_{10}$(adjusted *p*-value) followed by the BY multiple testing adjustment procedure. Left: gene set enrichment analysis results on T cells. Right: gene set enrichment analysis results on plasma cells. Source data are provided as a Source Data file.

the lower-left corner, whereas both utSVGs and ctSVGs preserved this critical information. These findings highlight that including ctSVGs significantly enhances domain detection accuracy and demonstrates the value of STANCE in facilitating this process. Similar conclusions can be drawn when using the other four domain detection algorithms (see Supplementary File Fig. S12).

For each gene, we estimated the cell-type-specific spatial effects. We plotted the top 20 ctSVGs to show the relative proportion of variances explained by cell type-specific effects and the error variance through stacked bar plots for each of the 8 cell types (Supplementary File Fig. S13). Among endothelial cells, significant ctSVGS included genes such as *RAMP3*, *PLVAP*, *CLDN5*, *VWF*, *EGFL7*, *ENG*, *AQP1*, *ICAM1*, and *ACVRL1*, pivotal for endothelial function. Notably, *RAMP3* regulates vascular tone and angiogenesis by interacting with G protein-coupled receptors[22], while *PLVAP* is crucial in fenestrated endothelia for diaphragm formation[23]. In plasma cells, genes like *IGHA2*, *IGLV3-21*, *IGHA1*, *IGLC3*, *IGKC*, *IGHG1*, *IGLC2*, and *MZB1* are essential for antibody production and immunoglobulin secretion (Fig. 6e). Additionally, the top gene *IL10RA* in T cells (Fig. 6e), expressed on the surface of T cells and other immune cells, is involved in signaling pathways regulating immune responses[24], while *SPINT2* (serine peptidase inhibitor, Kunitz type, 2), the top ctSVG of epithelial cells, is expressed in epithelial cells and functions as an inhibitor of serine proteases, playing a role in maintaining epithelial integrity[25]. Furthermore, other cell types show strong correlations between specific genes and their functions: *CD37* and *TCIRG1* in B cells, *COL6A3*, *POSTN*, *AEBP1*, *OLFML3*, *FN1*, and *MXRA8* in cancer-associated fibroblasts (CAFs), *FOXM1*, *TACSTD2*, *RXRA*, *TPD52*, and *CDH1* in epithelial cells, *TYROBP*, *IFI27*, *C1QB*, *CD14*, and *PFKL* in myeloid cells, and *CSPG4*, *SLPI*, and *ERP29* in perivascular-like (PVL) cells. These identified genes may serve as valuable marker genes with cell type-specific spatial patterns to further enhance domain detection in downstream analyses.

For the top ctSVG of each cell type, we generated spatial pattern plots of their scaled expression and outlined spots with colors based on the domain annotation (Supplementary File Fig. S14). The top ctSVG for T cells, *IL10RA*, is highly expressed in the region of cancer in situ in HER2+ breast tumors (Fig. 6d). The *IL10RA* gene encodes the alpha subunit of the receptor for interleukin-10 (IL-10), a cytokine involved in anti-inflammatory responses and immune regulation. IL-10 and its receptor are crucial in modulating immune responses by limiting inflammation and promoting immune tolerance. This mechanism could help tumor cells remain undetected and confined to their site of origin, explaining the high expression of *IL10RA* in the cancer in situ region[26]. Additionally, the *IGHA2* gene, which is the top ctSVG for plasma cells and encodes the heavy chain constant region of IgA2 (one of the two subclasses of immunoglobulin A), shows higher expression in the domain of breast glands compared to other domains (Fig. 6d). In normal breast tissue, especially within the glands, *IGHA2* expression is expected due to its role in local immunity and secretion into breast milk. However, in the context of a HER2+ breast tumor, the expression patterns of many genes, including *IGHA2*, might be altered due to changes in the local immune microenvironment and the overall disruption of normal tissue architecture and function caused by the tumor[27]. These findings highlight the necessity of ctSVG detection for understanding transcriptomic mechanisms through both spatial and

cell-type information. They also demonstrate the effectiveness of STANCE in detecting ctSVGs.

We also conducted gene set enrichment analysis to assess the significance of pathways associated with each group of ctSVGs. For each cell type, we identified its top 10 pathways, most of which deemed particularly meaningful (Supplementary File Fig. S15). Among plasma cells, specialized B cells responsible for producing antibodies, some pathways like "B cell receptor signaling pathway", "immunoglobulin mediated immune response", "B cell mediated immunity", and "humoral immune response mediated by circulating immunoglobulin" (Fig. 6f) are notably specific, while some other pathways like "complement activation (both classical pathway and general)" and "antimicrobial humoral response" are not exclusive to plasma cells; they are closely associated with plasma cells due to their role in antibody-mediated immune responses and complement system activation[28–34]. In B cells, all top 10 pathways identified are highly relevant to this cell type, given their crucial roles in the immune system. Specifically, the third pathway "adaptive immune response based on somatic recombination of immune receptors built from immunoglobulin superfamily domains" refers to the process by which B cell receptors (BCRs), which are immunoglobulins, are generated through somatic recombination of gene segments. This process is unique to B cells and results in a diverse repertoire of BCRs capable of recognizing a wide array of antigens[35]. The significant pathways enriching ctSVGS, detected by STANCE, suggest promising future applications in diagnostic approaches for breast cancer.

### Real data analysis: human kidney cancer data

We examined the 10x Visium spatial transcriptomics dataset from human kidney cancer tumor core[36], containing expression count data for 36,601 genes across 3008 spots. We used CARD, a reference-based cell-type deconvolution tool, to obtain cell-type composition information. Specifically, we followed the default quality control procedure of CARD to remove low-expressed genes and spots. We obtained the cell type proportions for 12 cell types, including B cells, plasma cells, T cells, natural killer (NK) cells, endothelial cells (EC), renal cell carcinoma (RCC) cells, non-proximal tubule epithelial (Epi_non-PT) cells, proximal tubule epithelial (Epi_PT) cells, fibrobalast cells, myeloid cells, plasmacytoid dendritic cells (pDC), as well as mast cells, across 2917 spots. We then removed mitochondrial genes and low-expressed genes that do not express at more than 10% spots, resulting in 7270 genes.

The STANCE overall test identified 4158 genes, with p-values adjusted by the Benjamini–Yekutieli method to meet an FDR of $p = 0.01$. These genes are considered utSVGs, a mixture of SVGs and ctSVGs. SPARK-G identified 4011 SVGs, 3781 of which were also identified by the STANCE overall test. SPARK-X identified 6292 SVGs, with 4067 overlapping with those detected by STANCE (Supplementary File Fig. S16).

For the 4158 utSVGs detected by the STANCE overall test, we conducted STANCE cell-type-specific individual test for each of the 12 cell types and identified a total of 2094 ctSVGs (at the 0.01 significance level) (Fig. 7a). Fibrobalast cells had the most ctSVGs with 676 genes, while plasma cells had the fewest, with 67 genes. Figure 7c displays the spatial expression pattern of two example genes *DCUN1D4* in pDC and

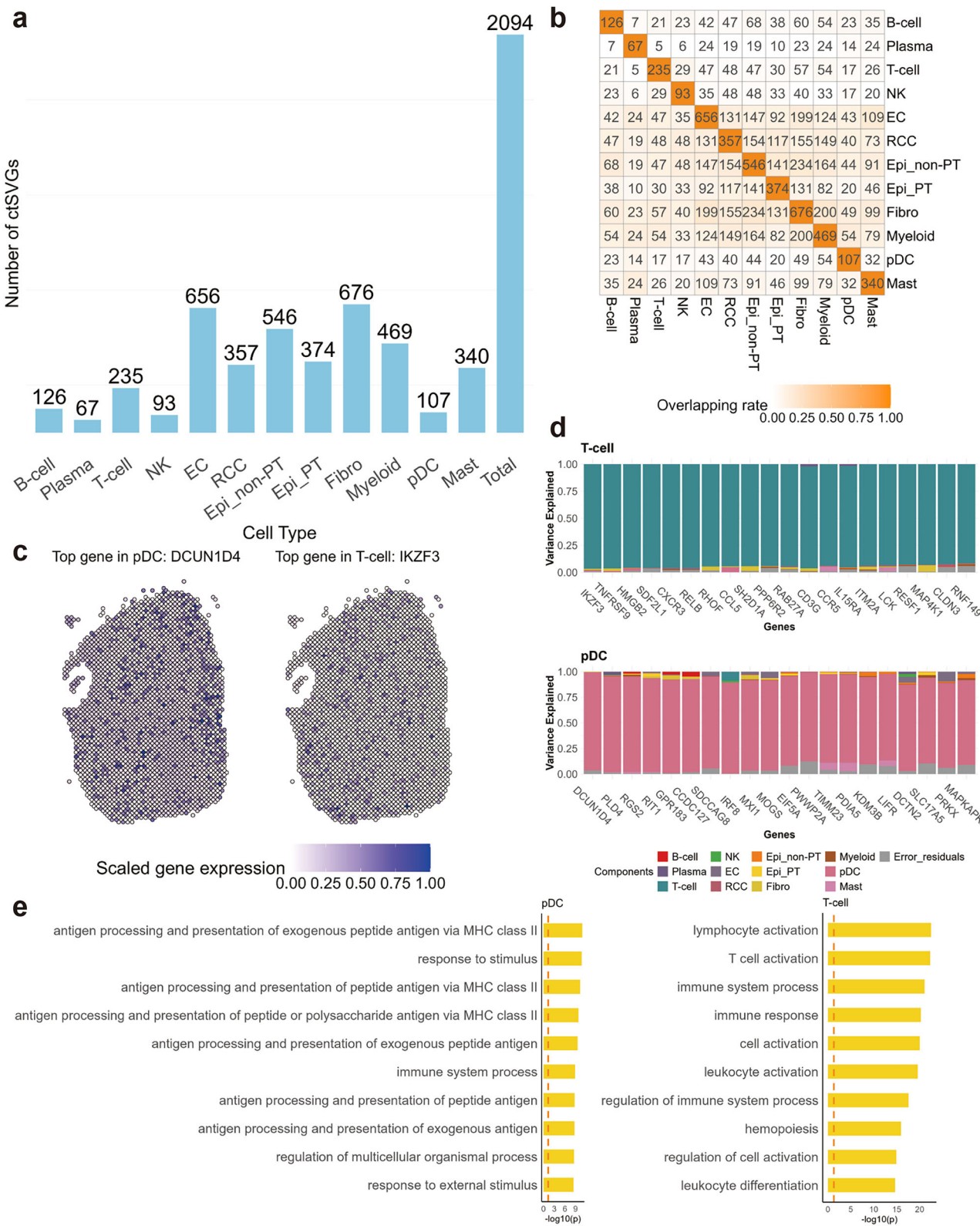

**Fig. 7 | Results of the human kidney cancer tumor core data. a** The bar plot displays the numbers of ctSVGs for each cell type. **b** The heatmap displaying the intersections of ctSVGs between cell types. **c** Spatial pattern plots of scaled gene expression for two example ctSVGs. Spots are outlined with colors indicating the domain annotation. Left: *DCUN1D4*. Right: *IKZF3*. **d** The stacked bar plots display the variance explained by 12 cell-type-specific spatial effects and random error, illustrating the top 20 ctSVGs for each cell type. Top: T cells. Bottom: plasmacytoid

dendritic cells (pDC). **e** The gene set enrichment analysis results on two groups of ctSVGs detected by STANCE. The top 10 significant pathways are shown. The enrichment is given as $-\log_{10}$(adjusted *p*-value) followed by the BY multiple testing adjustment procedure. Left: gene set enrichment analysis results on pDC. Right: gene set enrichment analysis results on T cells. Source data are provided as a Source Data file.

*IKZF3* in T-cell. The plots of other top genes in each cell type are shown in Supplementary File Fig. S19.

spVC was also implemented on this dataset as a compared method, which identified 95 ctSVGs under the original tissue pattern, 94 of which overlapped with those identified by STANCE. After a 30° rotation of the tissue, spVC detected 104 ctSVGs, including 86 that overlapped with those from the original pattern. Following a 60° rotation, spVC identified 110 ctSVGs, with 86 shared with the original pattern and 91 overlapping with the 30° rotated pattern (Supplementary File Fig. S17). The inconsistent ctSVGs detected under different spatial rotations by spVC highlight a potential issue with treating spatial effects as fixed, suggesting that such an approach should be avoided in practical applications.

We further analyzed the 2094 ctSVGs detected by STANCE. We assessed the variance explained by the 12 cell-type-specific spatial effects and random error, visualizing the top 20 ctSVGs with the highest variance explained through stacked bar plots for each of the 12 cell types (Supplementary File Fig. S18). Among the top 20 ctSVGS in myeloid cells, notable genes such as *MAFB*, *CD163*, *CPVL*, *C3AR1*, *PLXNC1*, *HLA-DOA*, *CEBPA*, *CD53*, *LY96*, *MS4A7*, and *ADA2* are known for their significant roles in myeloid cell function, differentiation, and immune response. Similarly, in pDCs, genes like *IRF8*, *MXI1*, *MOGS*, *EIF5A*, *IRF7*, *TIMM23*, *PDIA5*, *KDM3B*, *LIFR*, and *SLC17A5*, among others, play crucial roles in development, function, and immune response activities (Fig. 7d). T cells exhibit high expression or strong correlation with genes such as *TNFRSF9*, *CXCR3*, *CCL5*, *SH2D1A*, *CD3G*, *CCR5*, *IL15RA*, *ITM2A*, *LCK*, and *MAP4K1*, influencing T cell activation, signaling, and migration (Fig. 7d). Furthermore, the top gene *XBP1* in B cells is involved in B cell differentiation into plasma cells and the unfolded protein response (UPR) during plasma cell differentiation[37], while *DEFB1*, the top ctSVG of RCC, is down-regulated and linked to tumor progression[38]. Additionally, other cell types show strong correlations between specific genes and their functions: *CD37* in B cells, *FGR* in NK cells, *ENPP2*, *PTPRB*, *PLXDC1*, and *F2R* in endothelial cells, *FXYD2*, *SLCO4C1*, *PHKA2*, and *TCTA* in kidney epithelial cells, *SLC47A1*, *CDHR5*, and *NAT8* in proximal tubule epithelial cells, *PTGS1*, *LAMA1*, *FGFR4*, *IL4I1*, and *SERPINF2* in fibroblast cells, and *SP3*, *NDUFS3*, *HBB*, *OCRL*, and *PRPF38A* in mast cells. These identified genes serve as valuable marker genes with cell-type-specific spatial patterns for enhancing domain detection in further analyses.

We conducted gene set enrichment analysis to assess the significance of pathways associated with each group of ctSVGs. For each cell type, we identified its top 10 pathways, with 3 to 10 pathways per cell type deemed particularly meaningful (Supplementary File Fig. S20). In T cells, pathways such as "lymphocyte activation", "T cell activation", "immune system process", "immune response", "cell activation", "leukocyte activation", "regulation of immune system process", "regulation of cell activation", and "leukocyte differentiation" were predominant (Fig. 7e), highlighting their crucial roles in immune system functions and responses[39]. Among pDCs, specialized in antiviral immunity and immune regulation, pathways such as "antigen processing and presentation of exogenous peptide antigen via MHC class II" and "antigen processing and presentation of peptide antigen via MHC class II" were notably specific (Fig. 7e). These pathways underscored pDCs' unique capabilities in antigen presentation, crucial for their role in immune regulation and response. Additional pathways related to antigen processing, such as "antigen processing and presentation of peptide or polysaccharide antigen via MHC class II", were also prominent, reflecting pDCs' function in capturing and presenting antigens to T cells via MHC class II molecules[40,41].

Furthermore, the pathway "anatomical structure morphogenesis" emerged prominently for fibroblast cells, highlighting their pivotal role in the morphogenesis and remodeling of connective tissues and extracellular matrix components. This function is essential for tissue development, repair, and maintenance, emphasizing the structural

integrity and organization contributed by fibroblasts[42]. The significant pathways enriched by ctSVGS, detected by STANCE, suggest promising future applications in diagnostic approaches of kidney cancer.

### Real data analysis: mouse olfactory bulb data
The third data set we examined was the 100 $\mu m^2$ resolution spatial transcriptomics data of the mouse olfactory bulb (MOB)[43]. There are a total of 12 tissue samples from the same subject. The spatial orientations of the 12 samples are all different, highlighting the importance of analyzing ST data with spatial rotation invariant methods. We focused on MOB replicate #8, which consists of expression count data for 15,928 genes across 262 pixels (spots). We then deconvolved the MOB data using STdeconvolve[44], a reference-free and unsupervised cell-type deconvolution tool. Before cell type deconvolution, we followed the procedure of the original article of STdeconvolve to clean the dataset, resulting in 7365 genes and 260 spots. With STdeconvolve, 12 cell types were identified with their compositions at each spot. We also made a heatmap to visualize the transcriptional correlation between deconvolved cell types and cell clusters (Supplementary File Fig. S22). According to the heatmap, deconvolved cell type 4 and granular cell layer, deconvolved cell type 11 and mitral cell layer, deconvolved cell type 8 and outer plexiform layer, deconvolved cell type 2 and glomerular layer, and deconvolved cell type 12 and olfactory layer, are highly correlated with a correlation greater than 0.6.

We applied STANCE, SPARK-G, SPARK-X, and spVC to this dataset. The STANCE overall test identified 828 genes, with p-values adjusted by the Benjamini-Yekutieli method to meet an FDR of $p = 0.05$. These genes are considered utSVGs, a mixture of SVGs and ctSVGs. SPARK-G identified 188 SVGs, 187 of which were also identified by the STANCE overall test. SPARK-X identified 2,321 SVGs, with 447 overlapping with those detected by STANCE. In total, 127 genes were detected by all three methods (Supplementary File Fig. S21). Subsequently, we performed the STANCE cell type-specific individual test on the 828 genes identified by the overall test and analyzed each of the 12 cell types. In total, 809 genes were identified as ctSVGs specific to at least one cell type (Fig. 8a). Cell Type 12 had the most ctSVGs with 319 genes, while Cell Type 1 had the fewest, with 47 genes. Surprisingly, spVC was unable to identify any ctSVG on this dataset.

We evaluated the variance explained by the 12 cell type-specific spatial effects and random error, illustrating the top 20 ctSVG through stacked bar plots for each of the 12 cell types (Fig. 8c and Supplementary File Fig. S23). Furthermore, for the top ctSVG of each deconvolved cell type, we generated spatial pattern plots of their scaled expression and outlined spots with colors based on the cluster assignment associated with coarse cell layers (Supplementary File Fig. S24). Notably, the spatial patterns for the top ctSVGs of deconvolved cell types 2, 4, 9, 11, and 12 align with high transcriptional correlations (Supplementary File Fig. S22), suggesting that these top ctSVGs are likely marker genes for their respective cell types. This alignment demonstrates the effectiveness of STANCE in identifying cell type marker genes with non-random spatial patterns, reinforcing its potential utility in cell type deconvolution.

## Discussion
We have introduced STANCE, a unified statistical model developed to detect ctSVGs in spatial transcriptomics. By integrating gene expression, spatial location, and cell type composition through a linear mixed-effect model, STANCE enables the identification of both SVGs and ctSVGs in an initial stage, followed by a second stage dedicated to ctSVG detection. Its design ensures robustness in complex scenarios and maintains invariance to spatial coordinate rotation.

STANCE offers several distinct advantages over other SVG and ctSVG detection methods. Unlike other ctSVG approaches such as CTSV, C-SIDE, and spVC, STANCE is spatially rotation-invariant. When compared to popular SVG detection methods like SPARK and SPARK-

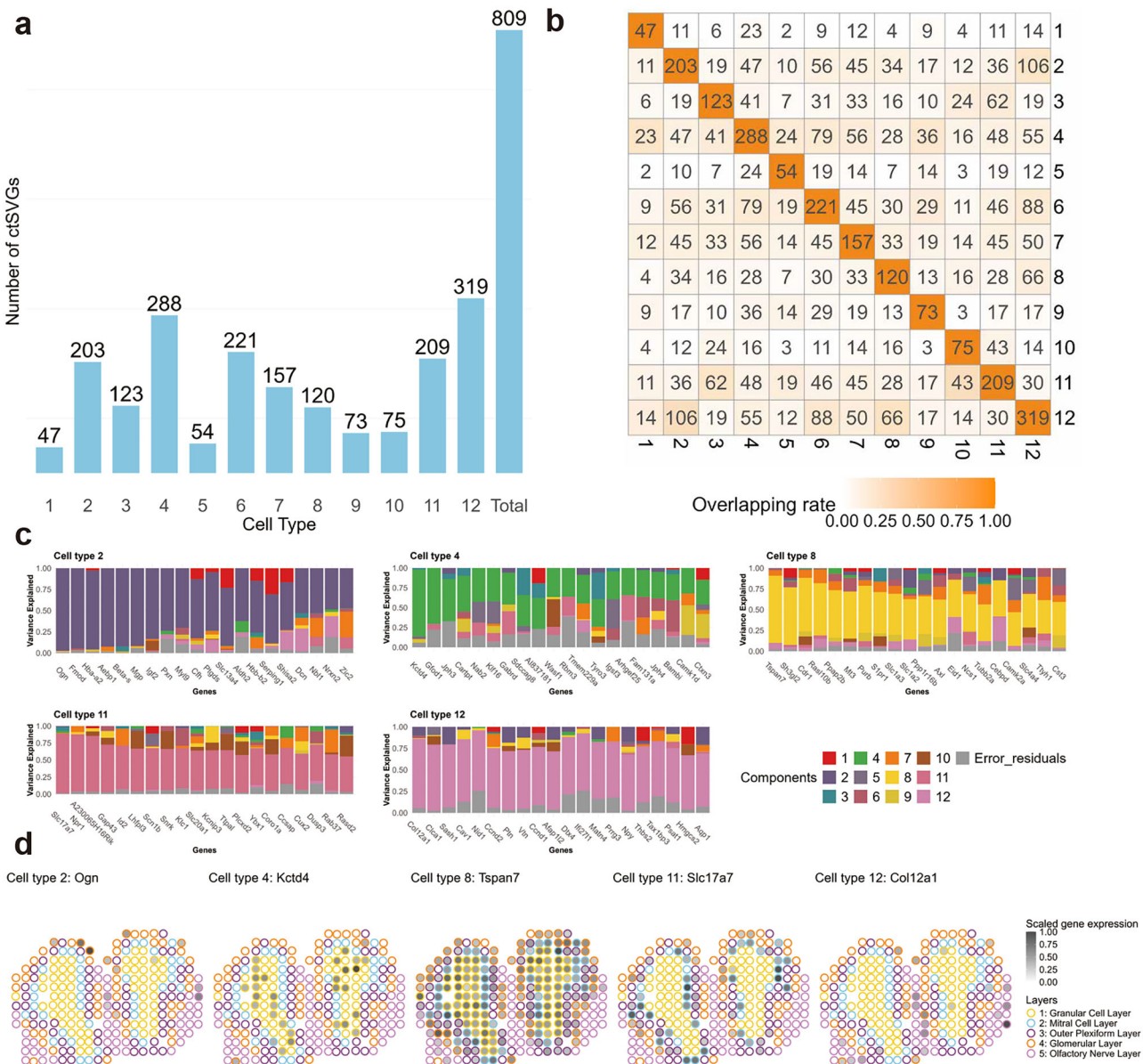

**Fig. 8 | Results of the mouse olfactory bulb data. a** The bar plot displays the numbers of ctSVGs for each cell type. **b** The heatmap displaying the intersections of ctSVGs between cell types. **c** The stacked bar plots display the variance explained by 12 cell type-specific spatial effects and random error, illustrating the top 20 ctSVGs for each of the 5 cell types with the highest matching with the layer annotation. Top left: deconvolved cell type 2. Top middle: deconvolved cell type 4. Top right: deconvolved cell type 8. Bottom left: deconvolved cell type 11. Bottom right: deconvolved cell type 12. **d** Spatial pattern plots of scaled gene expression for 5 example ctSVGs. Spots are outlined with colors indicating the layer annotation. Source data are provided as a Source Data file.

X, STANCE performs equally well or better in detecting SVGs, as evidenced by the results from Alternative Case 3 in our simulation studies. Moreover, STANCE uniquely provides the ability to estimate individual variance components, which are visualized through a stacked variance plot that displays the relative variance of a gene within each cell type. Additionally, utilizing utSVGs or ctSVGs identified by STANCE enhances spatial domain detection compared to SVGs detected by other methods such as SPARK, as demonstrated in the analysis of the Human breast cancer dataset. Finally, the downstream functional enrichment analysis further underscores the advantages of using STANCE.

STANCE can be easily extended to single-cell resolution ST data by modifying the $\pi_k$ vector. If the $i$th cell belongs to the $k$th cell type, then $\pi_{ik} = 1$, otherwise $\pi_{ik} = 0$. The same estimation and testing procedures can then be applied.

Despite its strengths, STANCE has areas for improvement. Firstly, it does not directly model raw count data, instead relying on a normalization procedure that may impact its power. We have created a generalized version of STANCE that models raw count data using a Poisson or negative binomial mixed-effect model. However, this increases the computational cost significantly. Therefore, we propose only the Gaussian version of STANCE to balance accuracy and computational efficiency. Future implementations may include techniques like low-rank approximation of kernels to enhance computational efficiency.

Additionally, STANCE's accuracy depends on the precision of cell-type deconvolution results. While STANCE is robust against mis-specifications in cell type composition for its type I error control, inaccurate estimates can affect its testing power in certain cases. For the Gaussian kernel, selecting an appropriate bandwidth parameter is crucial. For datasets with a small number of spots (e.g., fewer than 1000), one can adopt the approach used in SPARK by choosing a set of bandwidth parameters and aggregating the p-values across different

bandwidths to enhance power. For datasets with a large number of spots, the computational cost can be prohibitive. In such cases, applying a rule-of-thumb criterion[45,46] can save computational time, albeit with a trade-off in accuracy.

# Methods

## The model

Our goal is to identify genes that exhibit spatial expression patterns, referred to as utSVGs, and further test if they are specific to certain cell types (i.e., ctSVGs). Suppose we have spatial transcriptomics expression data for $q$ genes from $n$ spots (or pixels) of a 2D tissue, with their spatial locations denoted as $\mathbf{s} = (s_{i1}, s_{i2})_{n \times 2}$, $i = 1, \cdots, n$. The original gene expression count data of $n$ spots are collected and normalized through various methods to yield continuous gene expression data, denoted as $\mathbf{y} = (y_1, \cdots, y_n)^T$. Additionally, we assume that all the cells in the tissue belong to $K$ cell types, and for each spot, the cell type compositions have been estimated using existing cell type deconvolution tools (either reference-based or reference-free), denoted by $\mathbf{\Pi} = (\pi_{i1}, \cdots, \pi_{iK})_{n \times K}$, $i = 1, \cdots, n$.

With this information, we establish a variance component model[47] to elucidate the relationship between the spatial pattern of gene expression and cell type composition, i.e.,

$$\mathbf{y}(\mathbf{s}) = \mathbf{X}(\mathbf{s})\boldsymbol{\beta} + \boldsymbol{\gamma}(\mathbf{s}) + \boldsymbol{\varepsilon}(\mathbf{s}), \qquad (1)$$

where $\boldsymbol{\beta}$ are non-spatial fixed effects with $\mathbf{X}(\mathbf{s})$ being a $n \times p$-dimensional design matrix for covariates and $\boldsymbol{\beta} = (\beta_1, \cdots, \beta_p)^T$ being a $p$-dimensional vector of associated coefficients (when there is no covariate, $\mathbf{X}(\mathbf{s})$ contains elements of ones); $\boldsymbol{\gamma}(\mathbf{s})$ are the spatial random effects; $\boldsymbol{\varepsilon}(\mathbf{s}) = (\varepsilon_1, \cdots, \varepsilon_n)^T$ is a $n$-dimensional vector of random effects for residual errors, following a multivariate normal distribution, i.e., $MVN(\mathbf{0}, \sigma_\varepsilon^2 \mathbf{I}_n)$. We further decompose the spatial random effects $\boldsymbol{\gamma}(\mathbf{s})$ into $K$ cell-type-specific components, i.e.,

$$\boldsymbol{\gamma}(\mathbf{s}) = \boldsymbol{\pi}_1 \odot \boldsymbol{\gamma}_1(\mathbf{s}) + \cdots + \boldsymbol{\pi}_K \odot \boldsymbol{\gamma}_K(\mathbf{s}), \qquad (2)$$

where $\boldsymbol{\gamma}_k(\mathbf{s})$ is a $n$-dimensional vector of spatial random effects contributed by cell type $k$, $k = 1, \cdots, K$; $\boldsymbol{\pi}_k = (\pi_{1k}, \cdots, \pi_{nk})^T$ is the $k$-th column of $\mathbf{\Pi}$, whose $(i, k)$-th element $\pi_{ik}$ is the proportion of a specific cell type $k$ on spot $i$, $i = 1, \cdots, n$; $\odot$ is the Hamadard element-wise product. We assume that cell-type spatial random effects are independent of each other and each cell-type spatial random effect follows a multivariate normal distribution, i.e.,

$$\boldsymbol{\gamma}_k(\mathbf{s}) = (\gamma_k(\mathbf{s}_1), \cdots, \gamma_k(\mathbf{s}_n))^T \sim MVN(\mathbf{0}, \tau_k \mathbf{K}), k = 1, \cdots, K, \qquad (3)$$

where $\mathbf{K}$ is an $n \times n$ kernel matrix capturing the spatial similarity between spots; $\tau_k$ is the variance component of spatial effect corresponding to cell type $k$. Combining formulas (1) and (2), we obtain

$$\mathbf{y}(\mathbf{s}) = \mathbf{X}(\mathbf{s})\boldsymbol{\beta} + \boldsymbol{\pi}_1 \odot \boldsymbol{\gamma}_1(\mathbf{s}) + \cdots + \boldsymbol{\pi}_K \odot \boldsymbol{\gamma}_K(\mathbf{s}) + \boldsymbol{\varepsilon}(\mathbf{s}). \qquad (4)$$

which is the complete form of STANCE model. Then, the covariance matrix of $\mathbf{y}$ is given by

$$\mathrm{Cov}\,(\mathbf{y}) \equiv \mathbf{V} = \sum_{k=1}^{K} \tau_k \mathbf{\Pi}_k \mathbf{K} \mathbf{\Pi}_k^T + \sigma_\varepsilon^2 \mathbf{I}_n = \sum_{k=1}^{K} \tau_k \mathbf{\Sigma}_k + \sigma_\varepsilon^2 \mathbf{I}_n, \qquad (5)$$

where $\mathbf{\Pi}_k = \mathrm{diag}\{\boldsymbol{\pi}_k\}$ and $\mathbf{\Sigma}_k = \mathbf{\Pi}_k \mathbf{K} \mathbf{\Pi}_k^T$ for $k = 1, \cdots, K$.

STANCE incorporates spatial information by constructing a distance-based kernel matrix, denoted as $\mathbf{K}$, using spot coordinates. The $(i, j)$th-entry of this kernel matrix, $\mathbf{K}_{i,j}$, is expressed as a function of the Euclidean distance $\|\mathbf{s}_i - \mathbf{s}_j\|$ between the $i$-th and $j$-th spots, where $\mathbf{s}_i = (s_{i1}, s_{i2})^T$ and $\mathbf{s}_j = (s_{j1}, s_{j2})^T$ represent their respective coordinates, i.e., $\mathbf{K}_{i,j} = K(\|\mathbf{s}_i - \mathbf{s}_j\|)$. When the spatial coordinates are rotated by an angle $\theta$, the transformed coordinates of the $i$-th and $j$-th spots are given by $\tilde{\mathbf{s}}_i = \mathbf{R}\mathbf{s}_i$ and $\tilde{\mathbf{s}}_j = \mathbf{R}\mathbf{s}_j$, where the rotation matrix $\mathbf{R}$ can be defined as

$$\mathbf{R} = \begin{bmatrix} \cos\left(\frac{\theta}{180}\pi\right) & -\sin\left(\frac{\theta}{180}\pi\right) \\ \sin\left(\frac{\theta}{180}\pi\right) & \cos\left(\frac{\theta}{180}\pi\right) \end{bmatrix}$$

The matrix $\mathbf{R}$ is an orthogonal matrix, satisfying $\mathbf{R}^T\mathbf{R} = \mathbf{I}_2$, where $\mathbf{I}_2$ is a $2 \times 2$ identity matrix.

After rotation, the Euclidean distance between the $i$-th and $j$-th spots becomes

$$
\begin{aligned}
\| \tilde{\mathbf{s}}_i - \tilde{\mathbf{s}}_j \| &= \sqrt{(\tilde{\mathbf{s}}_i - \tilde{\mathbf{s}}_j)^T (\tilde{\mathbf{s}}_i - \tilde{\mathbf{s}}_j)} \\
&= \sqrt{(\mathbf{R}\mathbf{s}_i - \mathbf{R}\mathbf{s}_j)^T (\mathbf{R}\mathbf{s}_i - \mathbf{R}\mathbf{s}_j)} \\
&= \sqrt{(\mathbf{s}_i - \mathbf{s}_j)^T \mathbf{R}^T \mathbf{R} (\mathbf{s}_i - \mathbf{s}_j)} \\
&= \sqrt{(\mathbf{s}_i - \mathbf{s}_j)^T (\mathbf{s}_i - \mathbf{s}_j)} \\
&= \| \mathbf{s}_i - \mathbf{s}_j \|.
\end{aligned}
$$

The above calculation demonstrates that the Euclidean distance $\|\mathbf{s}_i - \mathbf{s}_j\|$ remains unchanged under rotation. Consequently, the distance-based kernel matrix $\mathbf{K}$ is invariant to various degrees of spatial rotations. Therefore, STANCE is theoretically rotation-invariant. Examples of rotation-invariant kernels are Gaussian kernel, Laplacian Kernel, and Matern Kernel.

## Hypothesis testing

Based on the STANCE model(4) with $K$ components representing the random effects on the spatial pattern of gene expression contributed by each of the $K$ cell types, we developed a unified two-stage testing procedure to systematically detect both spatially variable genes (SVGs) and cell type-specific spatially variable genes (ctSVGs).

First, we conduct an overall test for all the cell type-specific variance components with the hypotheses

$$\begin{cases} H_0^{(1)} : \tau_1 = \cdots = \tau_K = 0 \\ H_1^{(1)} : \text{at least one parameter is not zero} \end{cases} \qquad (6)$$

Under this null hypothesis, the model(4) simplifies to:

$$\mathbf{y}(\mathbf{s}) = \mathbf{X}(\mathbf{s})\boldsymbol{\beta} + \boldsymbol{\varepsilon}(\mathbf{s}),$$

which is a linear model with only fixed effects and an error term, indicating that the gene expression is unrelated to spatial locations. If the null hypothesis is rejected for a specific gene, it means that at least one $\tau_k \neq 0$, suggesting that at least one cell-type spatial effect contributes to the gene expression. This confirms the spatial variability of the gene, though the specific cell-type spatial variability remains undetermined. For the genes passing the stage 1 test, we call them utSVGs, which include both SVGs and ctSVGs. A score statistic is constructed for detecting utSVGs, and the corresponding p-value can be computed through approximation by a scaled chi-square distribution[48–51]. We examine one gene at a time, and once the p-values for all genes are obtained, a false discovery rate (FDR) control procedure (e.g., the Benjamini-Yekutieli procedure) is performed across all genes to declare the final significant gene list.

After obtaining the list of utSVGs, the next goal is to identify cell type-specific SVGs among them. Since the spatial effect of gene expression is decomposed into $K$ components, detecting the ctSVGs for a specific cell type $k$ involves testing whether its corresponding

variance component equals zero, that is,

$$\begin{cases} H_0^{(2)} : \tau_k = 0 \\ H_1^{(2)} : \tau_k > 0 \end{cases} \qquad (7)$$

Similarly, a score statistic is constructed to detect ctSVGs for specific cell types, and the associated p-value can also be computed through approximation by a scaled chi-square distribution[48–51]. For each cell type of interest, one gene is examined at a time. Finally, we obtain an $n \times K$ p-value matrix. The details about the estimation and testing can be found in Supplementary File.

## Simulation 1: evaluation of the overall test

Simulation 1 is to evaluate the performance of the STANCE overall test for both SVG and ctSVG detection. We first simulated single-cell resolution spatial transcriptomics data. We used a random-point-pattern Poisson process by utilizing the *rpoispp* function in the 'spat-stat' package to generate $N = 10,000$ cells randomly distributed within a unit square with each of them assigned precise spatial coordinates. Then, the assignment of these 10,000 cells to 3 spatial domains, denoted as $D1 - D3$ (Fig. 2a), is performed through the following procedure. All cells were initially categorized under domain $D1$. From the total $N$ cells, we randomly selected two without replacement to serve as the centers for domains $D2$ and $D3$. The radii for these domains were independently assigned by sampling from a uniform distribution between 0.1 and 0.5, introducing variability in domain sizes and influences. Cells were then reassigned to the closest domain center, based on the calculated radii, with an initial default to $D1$. In instances of overlapping domains, cells within such intersections were assigned to the domain with the smallest radius. This prioritizes more densely concentrated domains, potentially highlighting areas of higher biological significance.

We assumed the presence of 3 distinct cell types and simulated the expression of 1000 genes per cell using a series of negative binomial distributions characterized by specific mean and dispersion parameters. In this parameterization of negative binomial distribution, the variance is given by $\mu + \frac{\mu^2}{\phi}$. This relationship shows that the variance is directly influenced by the value of the dispersion parameter $\phi$. Specifically, as the dispersion parameter $\phi$ increases, the variance of the generated negative binomial random count numbers decreases. We developed several simulation cases to assess the type I error rates and testing power under scenarios of different cell type compositions and gene expression dispersion (with higher dispersion denoting smaller variation).

Spot-resolution spatial transcriptomics data were then simulated based on the single-cell resolution data generated from the procedure as described above. We created grids of size 0.03125 to split the unit square into $n = 1,024$ spots. For all cells inside each spot, we aggregated the expression counts to serve as the spot-level expression of this specific spot and also calculated the cell type compositions. Besides, the coordinates of each spot were constructed based on the means of the $x$ and $y$ coordinates of all the cells within this spot. The count data were then normalized for further analysis.

The null case. Each cell was assigned to one of three cell types based on a categorical distribution, with probabilities varying across three distinct scenarios (Fig. 2b). Specifically, in Scenario 1, the distribution was heavily skewed towards cell type 3, which had a 70% probability, while cell types 1 and 2 each had a 15% probability. In Scenario 2, the proportion for cell types 1, 2, and 3 was respectively set as 30%, 30% and 40%, respectively, while in Scenario 3, the proportion was set as 45%, 45% and 10%. The expression of 1000 genes was initially simulated using a series of negative binomial distributions, characterized by a mean of 1 and dispersion parameters of 0.7 and 1.5, and were subsequently normalized for further analysis. The spatial

patterns of cell type distributions under the three scenarios were displayed in Fig. 2c.

We selected 300 out of 1000 genes serving as cell type marker genes, in which each of the three cell types has 100 unique marker genes (for the deconvolution purpose). For each specific cell type, we set the expression of their marker genes with a fold change of 4 (i.e., multiplying the mean parameter of the negative binomial distribution by 4, regardless of their domain assignment). This resulted in the distribution of the 1000 genes into two categories: a cell-type marker gene group and a non-spatial gene group, with expected counts of 300 and 700 genes, respectively. The performance of the STANCE overall test in controlling the type I error was evaluated using the 700 genes in the non-spatial group. For each dispersion parameter and scenario, we simulated five replicates and combined the p-values across these replicates.

We also conducted a sensitivity test for the type I error control under the mis-specification of cell type compositions. Following the procedure described above, we replaced the true or estimated cell-type proportions with mis-specified ones. We assumed that the cell type composition of each spot was mis-specified by "a terrible deconvolution tool" to be a mixture of four cell types, with the proportions being four random numbers between 0 and 1 that sum to 1. In this scenario, either the number of cell types or the cell type proportions were mis-specified. Similar to the null case without mis-specification, we evaluated the performance of the STANCE overall test for type I error control on the 700 genes in the non-spatial group. For each dispersion parameter and scenario of true cell type composition, we simulated 5 replicates and combined the p-values across replicates.

Alternative case 1: utSVGs (both SVGs and CTSVGs). We followed the procedure of the null case to carry out cell type assignment and generated 300 cell type marker genes. Then, we assigned another 600 genes (serving as spatially variable genes) to three domains each with 200 unique domain-specific genes. For each specific domain, we modified the expression of the first 50 domain-specific genes with a fold change of 4 for cells located within this domain, regardless of their cell type, the next 50 genes with a fold change of 2, followed by 50 genes with a fold change of 0.5, and the last 50 domain-specific marker genes with a fold change of 0.25. Due to the distribution of cell types, these spatially variable genes are also cell type-specific spatially variable genes. The testing power was also evaluated under different fold changes. The spatial expression pattern of a representative utSVG is displayed in Fig. 2d and Supplementary File Fig. S4.

Consequently, the 1000 genes were categorized into three groups: cell type marker gene group (300), spatially variable gene group (600), and non-spatial gene group (100). The performance of the STANCE overall test to detect both SVGs and CSVGs is evaluated on the 600 genes in the spatial gene group. Similarly, for each dispersion parameter and each scenario, we simulated five replicates.

Alternative case 2: only ctSVGs. We followed the procedure of the null case to carry out cell type assignment and generated 300 cell type marker genes. Then, we treated the remaining 700 genes as cell type-specific spatially variable genes. We chose cell type 3 as the reference cell type when calculating the fold change. We picked two domains as spatially variable domains ($D1$ and $D2$). For cells in cell type 1, the mean expressions of these ctSVGs were two times higher in domain $D1$ than in $D3$ and were set to 0 in domain $D2$. On the contrary, for cells in cell type 2, the expressions of these ctSVGs were set to 0 in domain $D1$ and were two times higher in domain $D2$ than in $D3$. That is, the expressions of these genes are complementary in $D1$ and $D2$. Thus, for cell type 1 and 2, these 700 genes show cell type-specific spatial variation (hence ctSVGs), but for cell type 3 they do not show spatial variation (non-ctSVGs). When combining the three cell types together, these genes do not show spatial variation (thus non-SVGs). The spatial expression pattern of a representative ctSVG is displayed in Fig. 2d and Supplementary File Fig. S6.

The performance of the STANCE overall test to detect only ctSVGs is evaluated on the 700 ctSVGs. Similarly, for each dispersion parameter and each scenario, we simulated five replicates.

Alternative case 3: SVGs but not ctSVGs. Each cell was assigned to one of three cell types based on a categorical distribution, with probabilities varying across three distinct scenarios. In this case, the cell type compositions vary across domains, where each domain consists of only two cell types with equal proportions. That is, domain $D1$ consists of only cell types 1 and 2, domain $D2$ consists of only cell types 2 and 3, and domain $D3$ consists of only cell types 1 and 3. The expressions of 1000 genes were initially simulated using a series of negative binomial distributions, characterized by a mean of 1 and dispersion parameters of 0.7 and 1.5, like the other three cases above. The spatial expression pattern of a representative SVG is displayed in Fig. 2d and Supplementary File Fig. S7.

We selected 600 out of 1,000 genes serving as cell-type marker genes, in which each of the three cell types has 200 unique marker genes. For each specific cell type, we modified the expression of their first 50 marker genes with a fold change of 4 for cells in that cell type, regardless of their domain assignments, the next 50 marker genes with a fold change of 2, followed by 50 marker genes with a fold change of 0.5, and the rest 50 marker genes with a fold change of 0.25. As a result, the 1,000 genes were categorized into two groups: the cell-type marker gene group (600), and the non-spatial gene group (400). Due to the spatial distribution of cell types, the 600 cell-type marker genes display spatial patterns across all cell types, implying that they are SVGs but not ctSVGs. The performance of the STANCE overall test to detect only SVGs was evaluated on the 600 marker genes. For each dispersion parameter, we simulated five replicates.

## Simulation 2: evaluation of cell-type-specific test

Simulation 2 aims to evaluate the performance of the STANCE individual test in detecting ctSVGs. Similar to Simulation 1, we generated 4000 cells randomly distributed within a unit square, each assigned precise $x$ and $y$ coordinates. These cells were divided into two spatial domains: the spatially variable domain ($SD$) and the non-spatially variable domain ($D$) (Fig. 4c), following the procedure outlined below. First, one of the 4000 cells was randomly selected to serve as the center of $SD$. The radius of this domain was then independently sampled from a uniform distribution between 0.2 and 0.4. Any cell that fell within this circular area was assigned to the $SD$ domain, while the remaining cells were assigned to the $D$ domain.

Each cell was assigned to one of three cell types based on a categorical distribution, with probabilities of 10% for cell type 1, 30% for cell type 2, and 60% for cell type 3, reflecting low, medium, and high proportions, respectively (Fig. 4b). We simulated the expression of 1000 genes per cell using a series of negative binomial distributions with a mean of 1 and dispersion parameters of 0.7 and 1.5. Spot-resolution spatial transcriptomics data were then simulated based on the single-cell resolution data generated from the above procedure. The unit square was divided into 400 spots using a grid size of 0.05. For each spot, we aggregated the expression counts of all cells within it to determine spot-level expression and calculated cell type compositions. Additionally, the coordinates of each spot were based on the mean x and y coordinates of the cells within that spot.

Out of the 1000 genes, 300 were selected as cell-type marker genes, with 100 unique marker genes for each of the three cell types. We modified the expression of these marker genes with a fold change of 2 for their respective cell types, independent of domain properties. Next, we selected another 600 genes to serve as ctSVGs, with each of the three cell types having 200 unique ctSVGs. For example, for cell type 1, we adjusted the mean expression of its first 50 ctSVGs with a fold change of 4 for cells located within the spatially variable domain (SD), the next 50 ctSVGs with a fold change of 2, the following 50 ctSVGs with a fold change of 0.5, and the last 50 ctSVGs with a fold

change of 0.25. We applied a similar procedure to cell types 2 and 3, and simulated five replicates for each dispersion value to aggregate the p-values across replicates (Fig. 4a).

To evaluate the STANCE individual test's performance in controlling type I error, we combined the test results for detecting the other two cell types' specific SVGs in each cell type. Specifically, we combined the STANCE cell types 2 and 3 test results on cell type 1-specific genes, the STANCE cell types 1 and 3 test results on cell type 2-specific genes, and the STANCE cell types 1 and 2 test results on cell type 3-specific genes. These tests, which should not be rejected, were used to assess the type I error control.

The testing power of different methods were evaluated based on the testing results for detecting each cell type's ctSVGs. Figure 4d shows the spatial expression pattern of a cell type 3-specific ctSVG. Specifically, the test results on cell type 1-specific ctSVGs assess testing power under a low cell type proportion, the test results on cell type 2-specific ctSVGs assess power corresponding to a medium proportion, and the test results on cell type 3-specific ctSVGs assess power under a high proportion, with the results demonstrating the impact of cell type proportion on the testing power.

## Compared methods

Although the three methods (CTSV, C-SIDE and spVC) designed for ctSVG detection are not spatial rotation invariant and can lead to high false positives or false negatives, we still considered them in the context of Simulation 2 to provide a comprehensive comparison. However, practical constraints limited the inclusion of CTSV and C-SIDE in the final analysis. For CTSV, the required computational time was prohibitively long, making it impractical to include in the simulation comparison. For C-SIDE, it is tightly integrated with the deconvolution tool RCTD, which must be run prior to performing C-SIDE. However, in our simulation setting, RCTD filtered out a substantial percentage of ctSVGs, resulting in a failure to produce testing results for these genes. As a result, only spVC was included in Simulation 2. For this method, the full model was fitted on all simulated ctSVGs, and p-values for the spatially varying effects of three covariates (cell types) were collected. These p-values were subsequently combined for type I error rate control and power analysis, enabling a focused evaluation of spVC's performance in the simulation.

## Reporting summary

Further information on research design is available in the Nature Portfolio Reporting Summary linked to this article.

## Data availability

The human HER2+ breast cancer tumor dataset[13] can be found at https://zenodo.org. The human kidney cancer dataset can be found at https://data.mendeley.com. The mouse olfactory bulb (MOB) dataset[43] can be found at https://www.spatialresearch.org. Source data are provided with this paper.

## Code availability

The R code used to develop the model, perform the analyses and generate results in this study is publicly available on GitHub at https://github.com/Cui-STT-Lab/STANCE, under GPL-3.0 license. All citable codes of the present study[52] are publicly available at https://zenodo.org/records/14768010.

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

## Acknowledgements
This study was supported by a Strategic Partnership Grant from the Office of Research and Innovation and the high-performance computing center (HPCC) at Michigan State University.

## Author contributions
Y.C. conceived the idea. H.S. and Y.C. designed the experiments. H.S. developed the algorithm, implemented the software, performed simulations, and analyzed real data. Y.W. helped with the real data analysis. B.C. helped with data interpretation. H.S. and Y.C. wrote the manuscript.

## Competing interests
The authors declare no competing interests.
