## [Transparent Peer Review file · Nature Communications]

STANCE: a unified statistical model to detect cell-type-specific spatially variable genes in spatial transcriptomics

Corresponding Author: Professor Yuehua Cui

Version 0:

Reviewer comments:

Reviewer #1

(Remarks to the Author)

The manuscript introduces STANCE, a novel unified statistical model designed to detect both spatially variable genes (SVGs) and cell-type-specific spatially variable genes (ctSVGs) in spatial transcriptomics. STANCE is unique due to its spatial rotation invariance, a feature lacking in existing methods like CTSV, CSIDE, and spVC, which treat spatial effects as fixed and are therefore affected by changes in spatial orientation. I enjoyed reading this manuscript. However, there are some aspects I feel the authors should address further.

Major Comments

1. Rotation Invariance Justification: One important novelty of STANCE is that the method is invariant to rotation for SVG detection. However, there is limited evidence to well-support this motivation, which should be the focus of the STANCE method.

First, the necessity of having a method invariant to rotation regarding SVG detection is not well justified. Although the authors show the slice directions of the tissue MOB in the supplementary file, I suggest moving this information to the main text, along with the corresponding ctSVG (or SVG) detected by existing methods that are not invariant to rotation, to showcase how rotation might affect the discoveries as a teaser. Additionally, in the context of multi-slice spatial transcriptomics (SRT) data analysis where the slices are not matched, maintaining rotation invariance becomes crucial. Thus, a method for multi-slice integration while maintaining invariance to rotation is necessary. The authors could mention this scenario as a future direction.

Second, the manuscript dedicates limited space to demonstrating how existing ctSVG detection methods vary with slice rotation. I suggest adding numerical results regarding this aspect to highlight the technical gaps in the current literature.

2. Comparison with Existing ctSVG Detection Methods: Since in the existing numerical experiments implemented in STANCE no rotation was performed, it would be fair to compare STANCE with existing ctSVG detection methods. I am curious to see whether the discoveries made by existing methods differ from those of STANCE, and specifically, how the genes that repeatedly get selected in different rotations by existing methods differ from those detected by STANCE.

Minor Comment

1. Definition of 'utSVG': The term 'utSVG' is used without defining its full form. Please write out the full name when it is first mentioned to ensure clarity for the reader.

(Remarks on code availability)

The tutorial focuses on implementing the software STANCE. I wonder if the authors could also provide a tutorial on rotating existing data and illustrating STANCE's invariance to the rotation. This is important, especially for researchers who might wonder how rotation affects their downstream data analysis.

Reviewer #2

(Remarks to the Author)

In this article by Su et. al., the authors proposed a new method named STANCE, to identify spatially variable genes (SVGs) and cell type-specific spatially variable genes (ctSVGs) at increased precision. The authors proposed a linear mix-effect modeling approach, to address the challenges on the directionality and cell-type confounding effect. Overall, the manuscript presents a significant advancement in spatial transcriptome data analysis. My comments are below:

1. The authors could consider adding certain utility functions for the visualization of SVGs and ctSVGs in their package of STANCE.
2. The authors provided a thorough review of computational methodology evolution on ST data analysis, and ctSVGs detection algorithms. The descriptions were in details, with knowledge gaps in fixed effect modeling clearly demonstrated.
3. The term `utSVGs` in the last paragraph of the Introduction section was not defined.
4. In the description of STANCE, the stage 1 and stage 2 were described in the text but not clearly labelled in Figure 1. Thus, the authors should consider clearly identify stage 1 and 2 in the figure, to make the alignment.
5. In the simulation study under the null, seems that multiple methods did reasonably well. What is the possible reasoning for that? Additionally, the SPARK-X seems to show overly conservative performance, what's the possible reason?
6. In Figure 3, because the dispersion parameter plays a role in figure interpretation, I'd expect the authors could expand its concept a bit more in the main text.
7. In real data analysis, does the number of ctSVGs discovered also positively correlated with their cell-type proportions? If the goal is to discover ctSVGs for relatively rare cell types, would it be a good idea to use less stringent cutoff to determine significance?

(Remarks on code availability)

software package in good format and is ready for user.

Reviewer #3

(Remarks to the Author)

The authors propose a regression-based statistical model, STANCE, to detect cell-type-specific spatially variable genes in spatial transcriptomics. The proposed approach is a timely work targeting the rotation invariance challenges in spatial transcriptomics, and it uses a unified framework to identify SVGs and ctSVGs.

Major comments:

1. Rotation invariance is one of the biggest contributions of the work. However, the authors only demonstrate that existing works on CTSV, C-SIDE, and spVC do not have rotation invariance properties. How the proposed approach addresses this challenge is barely demonstrated in the manuscript, including the results of STANCE on this challenge, how it is designed to address the challenge, and why it works. It would be great to see the comparison of these non-invariance methods in comprehensive simulation and/or case studies.
2. In the simulation, experiments only use 30-degree rotation to demonstrate the failure of competitive methods. More in-depth descriptions and summaries with more rotation degrees are needed to show how severe this issue will affect the capacity of these methods and proposed methods. Another concern about this rotation simulation is that authors use RCTD to decompose and run the analysis. Then, they used RCTD again on the rotated samples and ran the analysis on the composition of the rotated cell type. It is more appropriate to run the analysis on one cell composition from RCTD, and the cell type composition should be invariant by its definition. Otherwise, it may be hard to distinguish whether the invariance comes from the RCTD method or the ctSVG analysis.
3. It is not crystal clear to me the definition of SVGs and ctSVGs. On stage 1, it targets 'both SVGs and ctSVGs' or 'either SVGs or ctSVGs'. Actually, SVGs are not necessarily ctSVGs, see Figure 2 in the CTSV paper (PMID: 35792822)
4. The rationale for identifying cell-type-specific SVGs other than cell-type-specific genes is not clear. For these identified ctSVGs, how they differ from the routine analysis without spatial information and how this spatial information provides better biological interpretation are not clear. It may be essential to compare it with a naive approach, saying get SVG first and use some test, e.g., Wilcoxon sum test, to identify cell-type-specific SVGs.
5. The simulation design is great, but it still needs some clarification. At the single-cell level, authors use 1000 genes, and 300 are selected as cell type markers. Will these 300 markers need to be adopted in the power analysis? They are also 'cell type-specific'. The authors set 600 ctSVGs vs 100 non-ctSVGs, this 6/1 ratio may be too high for practical usage and may cause problems in power analysis. At the spot level, authors have a four-fold change in the simulation, which may be too strong and will make it too easy for RCTD to get the same results as benchmarks. Fold change at two may make more sense. In addition, it is more appropriate to use zero-inflated negative binomial not negative binomial for the drop-out issue in sequencing.
6. It is not fair to make a comparison only with SPARK methods, which are not designed for ctSVGs. Existing ctSVGs identification methods need to be compared, even if they are not claimed to be rotation invariant.
7. The manuscript kept mentioning 'comparable to STANCE-oracle, indicating STANCE's robustness through deconvolution'. The authors need to justify how the extent of deconvolution correctness influences the results with comprehensive analysis and experiments, dummy STANCE-deconvolution to ground-truth deconvolution are both needed in the analysis. Similarly, three deconvolution methods (RCTD, CARD, STdeconvolve) are applied in three case studies, making it kind of cherry-picking in the process.
8. In domain identification, incorporating SVGs using only spatialPCA with results at ARI 0.39, 0.42, 0.45 may not sufficiently claim they significantly improve domain detection.
9. What is the difference in methodology between STANCE and Celina (https://lulushang.org/Celina_Tutorial/index.html)? Even though Celina has not been officially published, it could be interesting to see how they differ.
10. How about the computational time and memory usage of the STANCE? It may be helpful to test its capacity in cellular-

level sequencing data from 10X Xenium/Visium HD, et al. or directly state the analysis can be finished in seconds (minutes) on a desktop.

Minor comments:

1. In Figure 5B right, the legend says it is on fold change, but it looks like it is still at FDR like in Figure 5B left.
2. The existing method 'CSIDE' should be 'C-SIDE'.
3. The manuscript uses the terminology 'utSVG' as a mixture of SVGs and ctSVGs. This term should be defined at the first time it occurs at the end of the introduction.
4. In Section 4.3.1, the author mentioned "From the total N cells, we randomly selected three without replacement to serve as the centers for domains D2 and D3". Did the authors mean two points?

(Remarks on code availability)

I indeed install and run parts of the Rotation analysis on my windows machine but failed with SPARK on my Mac M1. The tutorial provided is kind of sufficient to me.

Suggestions: It may be better to provide required () function for installation, for some of the packages, such as SpatialExperiment and spVC, are not easy to install.

Reviewer #4

(Remarks to the Author)

(Remarks on code availability)

Version 1:

Reviewer comments:

Reviewer #1

(Remarks to the Author)

The authors have successfully addressed all my comments.

(Remarks on code availability)

The authors have updated the tutorials and functions according to the suggestions. The software should be suitable for public use.

Reviewer #2

(Remarks to the Author)

The authors have addressed all my questions.

(Remarks on code availability)

Software is user-ready.

Reviewer #3

(Remarks to the Author)

The authors significantly improved the quality of the manuscript, and they have already answered most of my questions. I only have one minor issue for my major review 2, so it would be better to show the results somewhere to demonstrate the invariance to the rotation when applying RCTD steps.

(Remarks on code availability)

The authors updated the codes according to my former recommendations

Reviewer #4

(Remarks to the Author)

(Remarks on code availability)

RE: NCOMMS-24-64397-T “STANCE: a unified statistical model to detect cell-type-specific spatially variable genes in spatial transcriptomics”

We greatly appreciate the insightful comments and suggestions from the reviewers. They helped us greatly improve the manuscript. Below we list our point-by-point responses to their comments. All the major changes in the manuscript are highlighted with red fonts.

Reviewer #1 (Remarks to the Author):

The manuscript introduces STANCE, a novel unified statistical model designed to detect both spatially variable genes (SVGs) and cell-type-specific spatially variable genes (ctSVGs) in spatial transcriptomics. STANCE is unique due to its spatial rotation invariance, a feature lacking in existing methods like CTSV, CSIDE, and spVC, which treat spatial effects as fixed and are therefore affected by changes in spatial orientation. I enjoyed reading this manuscript. However, there are some aspects I feel the authors should address further.

Thank you for your contribution to the review process. Our detailed responses are listed below.

Major Comments

1. **Rotation Invariance Justification:** One important novelty of STANCE is that the method is invariant to rotation for SVG detection. However, there is limited evidence to well-support this motivation, which should be the focus of the STANCE method.

First, the necessity of having a method invariant to rotation regarding SVG detection is not well justified. Although the authors show the slice directions of the tissue MOB in the supplementary file, I suggest moving this information to the main text, along with the corresponding ctSVG (or SVG) detected by existing methods that are not invariant to rotation, to showcase how rotation might affect the discoveries as a teaser. Additionally, in the context of multi-slice spatial transcriptomics (SRT) data analysis where the slices are not matched, maintaining rotation invariance becomes crucial. Thus, a method for multi-slice integration while maintaining invariance to rotation is necessary. The authors could mention this scenario as a future direction.

Thank you for your comments and suggestions! Based on your suggestion, we have added content to highlight the importance of rotation invariance in SRT analysis and to address the inconsistency of CTSV, C-SIDE, and spVC before and after spatial rotation. Below is a summary of the revisions made in response to your feedback:

1. Section “Introduction”

To emphasize the significance of rotation invariance, we revised the 8th paragraph of the "Introduction" section by incorporating the following statement:

“This issue is particularly problematic in the context of multi-slice spatial transcriptomics data analysis, where slices are not spatially aligned. Ensuring rotation invariance is critical for robust multi-slice integration. Consequently, treating spatial coordinates (and their transformations) as fixed effects is statistically unsound, highlighting the need for methods that account for rotation invariance when modeling cell type-specific spatial effects.”

2. Section “Results”

To clarify the inconsistency observed in existing methods, we added the following description as the second paragraph of the subsection "Method Overview and Simulations":

“We conducted a simulation study to evaluate the statistical robustness of existing methods, C-SIDE, spVC, and CTSV, in the detection of ctSVG (Simulation details are provided in Supplementary File I). These methods treat spatial locations as fixed effects, which compromises spatial rotation-invariance in their analyses. Our findings revealed inconsistencies in testing results for these methods before and after spatial rotations at various angles, despite the expectation that outcomes should remain unaffected by such rotations. Notably, testing results varied systematically with changes in the rotation angle. These inconsistencies underscore the unreliability of these methods for ctSVG detection, indicating their unsuitability for robust ctSVG analysis.”

3. Section “Methods”

To explain why STANCE is theoretically rotation-invariant, we added the following to the subsection "The Model" within the "Methods" section:

STANCE incorporates spatial information by constructing a distance-based kernel matrix, denoted as \mathbf{K} , using spot coordinates. The (i, j) -th-entry of this kernel matrix, $\mathbf{K}_{i,j}$, is expressed as a function of the Euclidean distance $\|\mathbf{s}_i - \mathbf{s}_j\|$ between the i -th and j -th spots, where $\mathbf{s}_i = (s_{i1}, s_{i2})^T$ and $\mathbf{s}_j = (s_{j1}, s_{j2})^T$ represent their respective coordinates, i.e., $\mathbf{K}_{i,j} = K(\|\mathbf{s}_i - \mathbf{s}_j\|)$. When the spatial coordinates are rotated by an angle θ , the transformed coordinates of the i -th and j -th spots are given by $\tilde{\mathbf{s}}_i = \mathbf{R}\mathbf{s}_i$ and $\tilde{\mathbf{s}}_j = \mathbf{R}\mathbf{s}_j$, where the rotation matrix \mathbf{R} can be defined as

$$\mathbf{R} = \begin{bmatrix} \cos(\frac{\theta}{180}\pi) & -\sin(\frac{\theta}{180}\pi) \\ \sin(\frac{\theta}{180}\pi) & \cos(\frac{\theta}{180}\pi) \end{bmatrix}$$

The matrix \mathbf{R} is an orthogonal matrix, satisfying $\mathbf{R}^T\mathbf{R} = \mathbf{I}_2$, where \mathbf{I}_2 is a 2×2 identity matrix.

After rotation, the Euclidean distance between the i -th and j -th spots becomes

$$\begin{aligned} \|\tilde{\mathbf{s}}_i - \tilde{\mathbf{s}}_j\| &= \sqrt{(\tilde{\mathbf{s}}_i - \tilde{\mathbf{s}}_j)^T(\tilde{\mathbf{s}}_i - \tilde{\mathbf{s}}_j)} \\ &= \sqrt{(\mathbf{R}\mathbf{s}_i - \mathbf{R}\mathbf{s}_j)^T(\mathbf{R}\mathbf{s}_i - \mathbf{R}\mathbf{s}_j)} \\ &= \sqrt{(\mathbf{s}_i - \mathbf{s}_j)^T\mathbf{R}^T\mathbf{R}(\mathbf{s}_i - \mathbf{s}_j)} \\ &= \sqrt{(\mathbf{s}_i - \mathbf{s}_j)^T(\mathbf{s}_i - \mathbf{s}_j)} \\ &= \|\mathbf{s}_i - \mathbf{s}_j\|. \end{aligned}$$

The above calculation demonstrates that the Euclidean distance $\|\mathbf{s}_i - \mathbf{s}_j\|$ remains unchanged under rotation. Consequently, the distance-based kernel matrix \mathbf{K} is invariant to various degrees of spatial rotations. Therefore, STANCE is theoretically rotation-invariant. Examples of rotation-invariant kernels are Gaussian kernel, Laplacian Kernel, and Matern Kernel.

Second, the manuscript dedicates limited space to demonstrating how existing ctSVG detection methods vary with slice rotation. I suggest adding numerical results regarding this aspect to highlight the technical gaps in the current literature.

Thanks for your suggestions! We added simulations with 60-degree and 90-degree rotations in Supplementary File I to demonstrate that CTSV, C-SIDE, and spVC produced different

testing results as the rotation angle varies. These inconsistent outcomes emphasize that these methods are unreliable for ctSVG detection.

2. Comparison with Existing ctSVG Detection Methods: Since in the existing numerical experiments implemented in STANCE no rotation was performed, it would be fair to compare STANCE with existing ctSVG detection methods. I am curious to see whether the discoveries made by existing methods differ from those of STANCE, and specifically, how the genes that repeatedly get selected in different rotations by existing methods differ from those detected by STANCE.

Thank you for your comment! Based on your suggestions, we considered CTSV, C-SIDE and spVC in the context of Simulation 2 to provide a comprehensive comparison and added a new subsection “Compared methods” in “Methods” section:

“Although the three methods (CTSV, C-SIDE and spVC) designed for ctSVG detection are not spatial rotation invariant and can lead to high false positives or false negatives, we still considered them in the context of Simulation 2 to provide a comprehensive comparison. However, practical constraints limited the inclusion of CTSV and C-SIDE in the final analysis. For CTSV, the required computational time was prohibitively long, making it impractical to include in the simulation comparison. For C-SIDE, it is tightly integrated with the deconvolution tool RCTD, which must be run prior to performing C-SIDE. However, in our simulation setting, RCTD filtered out a substantial percentage of ctSVGs, resulting in a failure to produce testing results for these genes. As a result, only spVC was included in Simulation 2. For this method, the full model was fitted on all simulated ctSVGs, and p-values for the spatially varying effects of three covariates (cell types) were collected. These p-values were subsequently combined for type I error rate control and power analysis, enabling a focused evaluation of spVC’s performance in the simulation.”

We also updated the Simulation 2 result figures as well as the associated descriptions in “Method overview and simulations” subsection as follows:

“Essentially, STANCE-oracle effectively controlled the type I error across two different dispersion settings (Figure 5a). When using the deconvolved cell type compositions through RCTD, STANCE-RCTD produced testing results comparable to STANCE-oracle that used true cell type compositions, indicating STANCE’s robustness under deconvolution. **In contrast, both spVC-oracle and spVC-RCTD exhibited slightly inflated p-values under true null scenarios, indicating weaker type I error control compared to STANCE. These findings highlight STANCE’s superior ability to maintain statistical rigor in type I error control. Additionally, the analysis revealed no significant differences in type I error control across the different dispersion settings, further showcasing the robustness of both STANCE configurations to variations in data dispersion.**

When detecting ctSVGs, STANCE-oracle exhibited strong testing power across a wide range of FDR levels. Although the power of STANCE-RCTD was slightly lower than that of STANCE-oracle due to the reduced accuracy of deconvolved cell type compositions, it still performed robustly. In comparison, STANCE consistently outperformed spVC in detecting ctSVGs across nearly all scenarios of data dispersion and cell type proportions. Notably, the power to detect ctSVGs increased as the proportion of the cell types corresponding to the ctSVGs increased (Figure 5b). This trend is reasonable, as higher proportions provide more information, thereby enhancing the detection capability. Additionally, we observed that

statistical power improved with higher dispersion settings (low variances), further underscoring the effectiveness of STANCE in various data conditions. These results collectively highlight the advantages of STANCE in detecting ctSVGs, particularly under diverse and challenging conditions.”

Moreover, for each real dataset, spVC was also applied as a comparative method given that some practical limitations prevented the inclusion of CTSV and C-SIDE in the analysis. Interestingly, spVC failed to identify any ctSVGs in human HER2-positive breast cancer tumor dataset and mouse olfactory bulb (MOB) dataset, as no genes passed its initial filtering step. This unexpected outcome suggests that spVC may exhibit excessive conservativeness in certain datasets or analysis scenarios, particularly during its initial screening process. Such stringency could limit its ability to detect ctSVGs under specific conditions, potentially overlooking biologically meaningful results.

As for the human kidney cancer tumor core dataset, we constructed a Venn diagram to illustrate the relationships between sets of ctSVGs identified by STANCE and spVC under various tissue rotation conditions (see section 2.2.2). spVC identified 95 ctSVGs under the original tissue pattern, 94 of which overlapped with those identified by STANCE. After a 30-degree rotation of the tissue, spVC detected 104 ctSVGs, including 86 that overlapped with those from the original pattern. Following a 60-degree rotation, spVC identified 110 ctSVGs, with 86 shared with the original pattern and 91 overlapping with the 30 degree rotated pattern. The inconsistent ctSVGs detected under different spatial rotations by spVC highlight a potential issue with treating spatial effects as fixed, suggesting that such approach should be avoided in practical applications. We have added this to the real data analysis section.

Minor Comment

1. Definition of 'utSVG': The term 'utSVG' is used without defining its full form. Please write out the full name when it is first mentioned to ensure clarity for the reader.

Thank you for your comment! Following your suggestion, we have updated the associated description for STANCE and utSVGs in the Introduction section as follows: “In the first stage, STANCE performs an overall test to identify the presence of SVGs and ctSVGs, classifying genes that pass this test as unified-type SVGs (utSVGs), which encompass both SVGs and

ctSVGs. These utSVGs are then subjected to a second stage analysis for ctSVG detection.”

Reviewer #1 (Remarks on code availability):

The tutorial focuses on implementing the software STANCE. I wonder if the authors could also provide a tutorial on rotating existing data and illustrating STANCE's invariance to the rotation. This is important, especially for researchers who might wonder how rotation affects their downstream data analysis.

Thank you for your comment! Following your suggestion, we have included a new function, “rotate_points()”, in Supplementary File I. This function applies an orthogonal transformation matrix R ($R^T R = I$) to the spatial coordinates, allowing for tissue rotation. As shown in Section 4.1 (on page 15), the kernel matrix is invariant to spatial rotation, which leads to the same testing results for STANCE since STANCE essentially uses the same kernel matrix for estimation and testing before and after rotation. Thus, we did not include the results for STANCE before and after rotation

Additionally, we have added additional simulations in Supplementary File I to demonstrate that CTSV, C-CIDE STANCE is not rotation-variant for ctSVG detection under different angles of rotation (i.e., 30, 60, and 90), further highlighting the drawback of these methods for ctSVG detection.

Reviewer #2 (Remarks to the Author):

In this article by Su et. al., the authors proposed a new method named STANCE, to identify spatially variable genes (SVGs) and cell type-specific spatially variable genes (ctSVGs) at increased precision. The authors proposed a linear mix-effect modeling approach, to address the challenges on the directionality and cell-type confounding effect. Overall, the manuscript presents a significant advancement in spatial transcriptome data analysis. My comments are below:

Thank you for your positive comment about our work! Our detailed responses are listed below.

1. The authors could consider adding certain utility functions for the visualization of SVGs and ctSVGs in their package of STANCE.

Thank you for your comment! Following your suggestions, we added a new function “visualizeGenePattern”, to the STANCE package, and the updated version is now available on GitHub (<https://github.com/Cui-STT-Lab/STANCE.git>) This function enables users to visualize the expression levels of a specified gene across spatial locations within the tissue, providing an intuitive representation of spatial gene expression patterns.

For detailed user instructions and examples, please refer to the updated tutorial (<https://haroldsu.github.io/STANCE/tutorial.html>) and the package manual available on the GitHub repository. We hope this enhancement will further facilitate the analysis and interpretation of spatial transcriptomics data. Thank you for your valuable suggestion!

2. The authors provided a thorough review of computational methodology evolution on ST data analysis, and ctSVGs detection algorithms. The descriptions were in details, with knowledge gaps in fixed effect modeling clearly demonstrated.

Thank you for your positive comment!

3. The term `utSVGs` in the last paragraph of the Introduction section was not defined.

Thank you for your comment! Following your suggestion, we have updated the associated description for STANCE and utSVGs in the Introduction section as follows: “In the first stage, STANCE performs an overall test to identify the presence of SVGs and ctSVGs, classifying genes that pass this test as unified-type SVGs (utSVGs), which encompass both SVGs and ctSVGs. These utSVGs are then subjected to a second stage analysis for ctSVG detection.”

4. In the description of STANCE, the stage 1 and stage 2 were described in the text but not clearly labelled in Figure 1. Thus, the authors should consider clearly identify stage 1 and 2 in the figure, to make the alignment.

Thank you for your comment! Following your suggestion, we updated the description in Figure 1. The terms “STANCE test 1 (overall test)” and “STANCE test 2 (individual test)” have been revised to “STANCE stage 1 (overall test)” and “STANCE stage 2 (individual test)” to ensure consistency and alignment.

5. In the simulation study under the null, seems that multiple methods did reasonably well. What is the possible reasoning for that? Additionally, the SPARK-X seems to show overly conservative performance, what’s the possible reason?

Thank you for your comment! The simulation study under the null was designed to evaluate the type I error control of the methods. Type I error control is a fundamental requirement for any statistical testing method, so it is expected that most methods perform well under this null simulation scenario.

Among the methods evaluated, SPARK-X stands out as the only one utilizing a non-parametric framework. This design choice prioritizes computational efficiency, making it significantly faster. However, as is common in such frameworks, there is often a trade-off between efficiency and statistical power. This trade-off likely contributes to SPARK-X's more conservative performance compared to other methods.

6. In Figure 3, because the dispersion parameter plays a role in figure interpretation, I’d expect the authors could expand its concept a bit more in the main text.

Thank you for your comment! The negative binomial distribution can be parameterized by the mean parameter μ , and the dispersion parameter ϕ . In this parameterization, the variance of the distribution is given by $\mu + \frac{\mu^2}{\phi}$. This relationship shows that the variance is directly influenced by the value of the dispersion parameter ϕ . Specifically, as the dispersion parameter ϕ increases, the variance of the generated negative binomial random count numbers decreases. This feature highlights the flexibility of the negative binomial distribution in modeling overdispersed count data, where the variance exceeds the mean.

To expand this concept according to your suggestions, we added the following descriptions about this parameterization of negative binomial distribution into the second paragraph of subsection “Simulation 1: Evaluation of the overall test” in the “Methods” section:

“In this parameterization of negative binomial distribution, the variance is given by $\mu + \frac{\mu^2}{\phi}$.”

This relationship shows that the variance is directly influenced by the value of the dispersion parameter ϕ . Specifically, as the dispersion parameter ϕ increases, the variance of the generated negative binomial random count numbers decreases.”

Additionally, we added the following statements on page 6 in the subsection “Method and simulations” in the “Results” section:

“The power analysis results across all three alternative cases demonstrate a common trend: statistical power increases as the variance decreases (or the dispersion increases). This observation is biologically and statistically intuitive. Lower variance in gene expression reduces noise, making the differences in expression levels between distinct spatial domains more consistent and reliable. As a result, the underlying gene expression patterns become more pronounced and easier to detect. This enhanced stability allows for more robust identification of differentially expressed genes associated with specific spatial regions. In biological systems, reduced variability often reflects tightly regulated processes or well-defined spatial organization, which can further amplify the signal-to-noise ratio and improve the detectability of spatially variable genes. Thus, the observed relationship between dispersion, variance, and power aligns with expectations from both biological mechanisms and statistical principles.”

7. In real data analysis, does the number of ctSVGs discovered also positively correlated with their cell-type proportions? If the goal is to discover ctSVGs for relatively rare cell types, would it be a good idea to use less stringent cutoff to determine significance?

Thanks for the comment! We acknowledge the challenge in establishing a direct relationship between the number of ctSVGs detected and the overall proportion of each cell type across the tissue, as ST data do not provide global cell type proportions. However, intuitively, if a cell type is rare across the tissue, as discussed in the “Results” section, the power of STANCE’s individual tests would be affected. This is because the statistical power to detect ctSVGs is influenced by the proportion of cells expressing the relevant genes. A lower cell-type proportion reduces the signal-to-noise ratio, making it more difficult to detect the associated ctSVGs. Consequently, the number of ctSVGs identified for rare cell types is likely to decrease. The same challenge also applies to other ctSVG detection methods.

Regarding the use of different significance thresholds for detecting ctSVGs across cell types, we think this approach should be used with caution for several reasons. In statistical analysis, using different thresholds for significance undermines the ability to control the overall error rate consistently. This inconsistency can lead to inflated false-positive rates or overly conservative results, depending on the thresholds chosen. Moreover, varying significance levels between tests make the results incomparable. For instance, if one test uses a cutoff of $p < 0.01$ and another with $p < 0.05$, the relative evidence against the null hypothesis for these tests cannot be directly compared. This inconsistency complicates interpretation. To ensure a robust and interpretable analysis, it is essential to maintain a consistent significance threshold across all tests when detecting ctSVGs for different cell types. This approach ensures the reliability and comparability of results while controlling for error rates effectively.

Reviewer #2 (Remarks on code availability):

software package in good format and is ready for user.

Thank you for your positive feedback!

Reviewer #3 (Remarks to the Author):

The authors propose a regression-based statistical model, STANCE, to detect cell-type-specific spatially variable genes in spatial transcriptomics. The proposed approach is a timely work targeting the rotation invariance challenges in spatial transcriptomics, and it uses a unified framework to identify SVGs and ctSVGs.

Thank you for your contribution to the review process. Our detailed responses are listed below.

Major comments:

1. Rotation invariance is one of the biggest contributions of the work. However, the authors only demonstrate that existing works on CTSV, C-SIDE, and spVC do not have rotation invariance properties. How the proposed approach addresses this challenge is barely demonstrated in the manuscript, including the results of STANCE on this challenge, how it is designed to address the challenge, and why it works. It would be great to see the comparison of these non-invariance methods in comprehensive simulation and/or case studies.

Thank you for your comment! Following your suggestions, we have added a theoretical evaluation to show that STANCE has the spatial rotation-invariant property (see section 4.1 and below). In Supplementary File I, we added simulations under three different spatial rotation angles, namely 30, 60, and 90. The results demonstrate that CTSV, C-CIDE, and spVC generated inconsistent results under different rotation angles, showcasing that they are not spatial rotation-invariant. Under different rotation angles, STANCE always produced the same results, hence we did not show the STANCE results under different rotations.

STANCE incorporates spatial information by constructing a distance-based kernel matrix, denoted as \mathbf{K} , using spot coordinates. The (i, j) -th-entry of this kernel matrix, $\mathbf{K}_{i,j}$, is expressed as a function of the Euclidean distance $\|\mathbf{s}_i - \mathbf{s}_j\|$ between the i -th and j -th spots, where $\mathbf{s}_i = (s_{i1}, s_{i2})^T$ and $\mathbf{s}_j = (s_{j1}, s_{j2})^T$ represent their respective coordinates, i.e., $\mathbf{K}_{i,j} = K(\|\mathbf{s}_i - \mathbf{s}_j\|)$. When the spatial coordinates are rotated by an angle θ , the transformed coordinates of the i -th and j -th spots are given by $\tilde{\mathbf{s}}_i = \mathbf{R}\mathbf{s}_i$ and $\tilde{\mathbf{s}}_j = \mathbf{R}\mathbf{s}_j$, where the rotation matrix \mathbf{R} can be defined as

$$\mathbf{R} = \begin{bmatrix} \cos(\frac{\theta}{180}\pi) & -\sin(\frac{\theta}{180}\pi) \\ \sin(\frac{\theta}{180}\pi) & \cos(\frac{\theta}{180}\pi) \end{bmatrix}$$

The matrix \mathbf{R} is an orthogonal matrix, satisfying $\mathbf{R}^T\mathbf{R} = \mathbf{I}_2$, where \mathbf{I}_2 is a 2×2 identity matrix.

After rotation, the Euclidean distance between the i -th and j -th spots becomes

$$\begin{aligned} \|\tilde{\mathbf{s}}_i - \tilde{\mathbf{s}}_j\| &= \sqrt{(\tilde{\mathbf{s}}_i - \tilde{\mathbf{s}}_j)^T(\tilde{\mathbf{s}}_i - \tilde{\mathbf{s}}_j)} \\ &= \sqrt{(\mathbf{R}\mathbf{s}_i - \mathbf{R}\mathbf{s}_j)^T(\mathbf{R}\mathbf{s}_i - \mathbf{R}\mathbf{s}_j)} \\ &= \sqrt{(\mathbf{s}_i - \mathbf{s}_j)^T\mathbf{R}^T\mathbf{R}(\mathbf{s}_i - \mathbf{s}_j)} \\ &= \sqrt{(\mathbf{s}_i - \mathbf{s}_j)^T(\mathbf{s}_i - \mathbf{s}_j)} \\ &= \|\mathbf{s}_i - \mathbf{s}_j\|. \end{aligned}$$

The above calculation demonstrates that the Euclidean distance $\|s_i - s_j\|$ remains unchanged under rotation. Consequently, the distance-based kernel matrix \mathbf{K} is invariant to various degrees of spatial rotations. Therefore, STANCE is theoretically rotation-invariant. Examples of rotation-invariant kernels are Gaussian kernel, Laplacian Kernel, and Matern Kernel.

2. In the simulation, experiments only use 30-degree rotation to demonstrate the failure of competitive methods. More in-depth descriptions and summaries with more rotation degrees are needed to show how severe this issue will affect the capacity of these methods and proposed methods. Another concern about this rotation simulation is that authors use RCTD to decompose and run the analysis. Then, they used RCTD again on the rotated samples and ran the analysis on the composition of the rotated cell type. It is more appropriate to run the analysis on one cell composition from RCTD, and the cell type composition should be invariant by its definition. Otherwise, it may be hard to distinguish whether the invariance comes from the RCTD method or the ctSVG analysis.

Thank you for your comment! Following your suggestions, we added 60-degree and 90-degree rotations in Supplementary File I to demonstrate that CTSV, C-SIDE, and spVC produce different testing results as the rotation angle changes. In the real data analysis, we also applied different rotations and analyzed the data using spVC. The results also showed inconsistent results under different rotations (see section 2.2.2). These inconsistent outcomes emphasize that these methods are unreliable for ctSVG detection.

Regarding your other concern, as outlined in the C-SIDE tutorial, the “create.RCTD” and “run.RCTD” functions must be executed before performing C-SIDE. Therefore, we applied RCTD to both the original and rotated patterns. We verified that the estimated cell type compositions remain invariant to rotation by including an additional step for validation.

3. It is not crystal clear to me the definition of SVGs and ctSVGs. On stage 1, it targets 'both SVGs and ctSVGs' or 'either SVGs or ctSVGs'. Actually, SVGs are not necessarily ctSVGs, see Figure 2 in the CTSV paper (PMID: 35792822)

Yes, you are right. Based on Figure 2 in the CTSV paper, SVGs are not necessarily ctSVGs while ctSVGs are not necessarily SVGs. The STANCE stage 1 overall test targets both SVGs and ctSVGs, which means that both SVGs and ctSVGs can be detected by STANCE at Stage 1. We call such SVGs passing stage 1 test as unified-type SVGs (utSVGs). Stage 2 test is to further distinguish if an utSVG is ctSVG or just regular SVG.

4. The rationale for identifying cell-type-specific SVGs other than cell-type-specific genes is not clear. For these identified ctSVGs, how they differ from the routine analysis without spatial information and how this spatial information provides better biological interpretation are not clear. It may be essential to compare it with a naive approach, saying get SVG first and use some test, e.g., Wilcoxon sum test, to identify cell-type-specific SVGs.

Thank you for your comment. The rationale for identifying ctSVGs instead of just cell-type-specific genes or SVGs is rooted in the goal of integrating both spatial variation and cell type composition information during gene detection. This integrated approach enhances downstream analyses, such as domain detection and enrichment analysis, by allowing for a more nuanced understanding of how genes vary spatially across different cell types within the tissue. Identifying ctSVGs provides a more precise representation of spatially variable

gene expression that is specific to particular cell types, which is critical for accurately mapping spatial organization and identifying functional domains within the tissue. To further emphasize the importance of this approach, we have modified the third paragraph in the Introduction section as follows:

“In both sequencing- and imaging-based ST data, gene expression variation across spatial spots often arises from differences in expression levels among distinct cell types. Since cell types are unevenly distributed within tissues, disregarding cell type information can obscure spatial patterns that extend beyond those driven by cell-type composition. Moreover, many SVGs with spatial expression variation are closely associated with cell type categories or compositions, giving rise to the concept of cell-type-specific spatially variable genes (ctSVGs). ctSVGs are characterized by non-random spatial expression patterns within specific cell types. Notably, an SVG may exhibit random spatial patterns within individual cell types, and conversely, a ctSVG may appear spatially random when considered more broadly, emphasizing that an SVG and a ctSVG are not inherently interchangeable. This distinction highlights the necessity of specialized methods for ctSVG detection, which rely on integrating SRT data with external cell-type annotations for spatial spots.”

Regarding the approach you suggested, after careful consideration, we think it may be challenging to do it. The main difficulty lies in the absence of labels or reasonable criteria to divide the identified SVGs into two distinct groups for the Wilcoxon sum test. Unlike single-cell resolution transcriptomics data, where individual cells can be distinctly identified and classified, each spot in ST data represents a mixture of multiple cell types. This complexity makes it challenging to separate the SVGs into meaningful groups, as the spatial spots are not homogeneous cell populations. Therefore, applying the Wilcoxon sum test in this context may not lead to reliable or interpretable results. Additionally, as noted in Comment 4, an SVG is not necessarily a ctSVG, and vice versa. Therefore, testing for ctSVGs based solely on the claimed SVGs may overlook potential ctSVGs that do not qualify as SVGs.

5. The simulation design is great, but it still needs some clarification. At the single-cell level, authors use 1000 genes, and 300 are selected as cell type markers. Will these 300 markers need to be adopted in the power analysis? They are also ‘cell type-specific’.

Thank you for your comment! The 300 cell-type marker genes will not be included in the power analysis. This is because, under our simulation settings, these markers are cell-type-specific but exhibit random spatial patterns due to the uniform distribution of cell types within the tissue. Though they are cell type-specific, but are not ctSVGs. They are used for cell type deconvolution purpose (see the 2nd paragraph on page 18).

At the spot level, authors have a four-fold change in the simulation, which may be too strong and will make it too easy for RCTD to get the same results as benchmarks. Fold change at two may make more sense.

Following your suggestions, we adjusted the fold change of the 300 cell type marker genes to 2 in the setting of Simulation 2, which led to a decrease in the power of both STANCE-RCTD and spVC-RCTD. This outcome is reasonable, as a lower fold change may reduce the accuracy of cell type deconvolution, thereby impacting the detection of ctSVGs. The updated results are as follows, reflecting this adjustment. We also added a comparison with spVC which is developed for ctSVG detection.

We also updated the last two paragraphs of subsection “Method overview and simulations” to incorporate this change. Additionally, Figure 4 in Simulation 2 has been revised to incorporate this change for clarity and consistency.

The authors set 600 ctSVGs vs 100 non-ctSVGs, this 6/1 ratio may be too high for practical usage and may cause problems in power analysis.

Thanks for your comment! We conducted additional simulation with a ratio of 300 ctSVGs to 300 non-spatial genes (1:1) and replicated the experiment 10 times, also resulting in a total of 3000 p-values for power analysis as the original setting. The power plot is included as Figure S9 in Supplementary file II and listed below.

Under the ctSVG vs. non-spatial genes ratio of 1:1, the power of STANCE-RCTD is closer to that of STANCE-oracle compared to the original ratio setting. However, it remains challenging to determine which ratio setting is more favorable for RCTD deconvolution.

In addition, it is more appropriate to use zero-inflated negative binomial not negative binomial for the drop-out issue in sequencing.

Thanks for the comment! Zhao et al. (2022) conducted comprehensive evaluations for ST data modeling and concluded that zero-inflated models are unnecessary in ST data modeling. Following this work, we opted to use the negative binomial model to simulate count data, as it provides sufficient accuracy without necessitating the complexity of a zero-inflated negative binomial model.

Ref: Zhao, P., Zhu, J., Ma, Y. et al. Modeling zero inflation is not necessary for spatial transcriptomics. *Genome Biology* 23, 118 (2022). <https://doi.org/10.1186/s13059-022-02684-0>

6. It is not fair to make a comparison only with SPARK methods, which are not designed for ctSVGs. Existing ctSVGs identification methods need to be compared, even if they are not claimed to be rotation invariant.

Thank you for your comment! Based on your suggestions, we considered CTSV, C-SIDE and spVC in the context of Simulation 2 to provide a comprehensive comparison and added a new subsection “Compared methods” in “Methods” section:

“Although the three methods (CTSV, C-SIDE and spVC) designed for ctSVG detection are not spatial rotation invariant and can lead to high false positives or false negatives, we still considered them in the context of Simulation 2 to provide a comprehensive comparison. However, practical constraints limited the inclusion of CTSV and C-SIDE in the final analysis. For CTSV, the required computational time was prohibitively long, making it impractical to include in the simulation comparison. For C-SIDE, it is tightly integrated with the deconvolution tool RCTD, which must be run prior to performing C-SIDE. However, in our

simulation setting, RCTD filtered out a substantial percentage of ctSVGs, resulting in a failure to produce testing results for these genes. As a result, only spVC was included in Simulation 2. For this method, the full model was fitted on all simulated ctSVGs, and p-values for the spatially varying effects of three covariates (cell types) were collected. These p-values were subsequently combined for type I error rate control and power analysis, enabling a focused evaluation of spVC's performance in the simulation."

We also updated the Simulation 2 result Figure 5 (see below as well). The results indicate that spVC has an inflated type I error and the power of spVC is lower than STANCE under different scenarios.:

The associated revision is further explained in the last two paragraphs in Section 2.1.

"Essentially, STANCE-oracle effectively controlled the type I error across two different dispersion settings (Figure 5a). When using the deconvolved cell type compositions through RCTD, STANCE-RCTD produced testing results comparable to STANCE-oracle that used true cell type compositions, indicating STANCE's robustness under deconvolution. In contrast, both spVC-oracle and spVC-RCTD exhibited slightly inflated p-values under true null scenarios, indicating weaker type I error control compared to STANCE. These findings highlight STANCE's superior ability to maintain statistical rigor in type I error control. Additionally, the analysis revealed no significant differences in type I error control across the different dispersion

settings, further showcasing the robustness of both STANCE configurations to variations in data dispersion.

When detecting ctSVGs, STANCE-oracle exhibited strong testing power across a wide range of FDR levels. Although the power of STANCE-RCTD was slightly lower than that of STANCE-oracle due to the reduced accuracy of deconvolved cell type compositions, it still performed robustly. In comparison, STANCE consistently outperformed spVC in detecting ctSVGs across nearly all scenarios of data dispersion and cell type proportions. Notably, the power to detect ctSVGs increased as the proportion of the cell types corresponding to the ctSVGs increased (Figure 5b). This trend is reasonable, as higher proportions provide more information, thereby enhancing the detection capability. Additionally, we observed that statistical power improved with higher dispersion settings (low variances), further underscoring the effectiveness of STANCE in various data conditions. These results collectively highlight the advantages of STANCE in detecting ctSVGs, particularly under diverse and challenging conditions.”

Moreover, for each real dataset, spVC was also applied as a comparative method given that some practical limitations prevented the inclusion of CTSV and C-SIDE in the analysis. Interestingly, spVC failed to identify any ctSVGs in the human HER2-positive breast cancer tumor dataset and mouse olfactory bulb (MOB) dataset, as no genes passed its initial filtering step. This unexpected outcome suggests that spVC may exhibit excessive conservativeness in certain datasets or analysis scenarios, particularly during its initial screening process. Such stringency could limit its ability to detect ctSVGs under specific conditions, potentially overlooking biologically meaningful results.

As for the human kidney cancer tumor core dataset, we constructed a Venn diagram to illustrate the relationships between sets of ctSVGs identified by STANCE and spVC under various tissue rotation conditions (see Section 2.2.2 and Figure S16 in Supplemental File II). spVC identified 95 ctSVGs under the original tissue pattern, 94 of which overlapped with those identified by STANCE. After a 30-degree rotation of the tissue, spVC detected 104 ctSVGs, including 86 that overlapped with those from the original pattern. Following a 60-degree rotation, spVC identified 110 ctSVGs, with 86 shared with the original pattern and 91 overlapping with the 30-degree rotated pattern. The inconsistent ctSVGs detected under different spatial rotations by spVC highlight a potential issue with treating spatial effects as fixed, suggesting that such an approach should be avoided in practical applications. We have added this to the real data analysis section.

7. The manuscript kept mentioning 'comparable to STANCE-oracle, indicating STANCE's robustness through deconvolution'. The authors need to justify how the extent of deconvolution correctness influences the results with comprehensive analysis and experiments, dummy STANCE-deconvolution to ground-truth deconvolution are both needed in the analysis.

Thank you for your comment! To demonstrate the robustness of STANCE to deconvolution results, we designed a sensitivity analysis for the type I error control under the misspecification of cell type compositions. We assumed that the cell type composition of each spot was mis-specified by "a terrible deconvolution tool" to be a mixture of four cell types, with the proportions being four random numbers between 0 and 1 that sum to 1 (the truth is 3 cell types). In this scenario, both the number of cell types and the cell type proportions were misspecified. Then, we evaluated the performance of the STANCE overall test for type I error control on the 700 genes in the non-spatial group. For each dispersion parameter and scenario of true cell type composition, we simulated 5 replicates and combined the p-values across replicates to evaluate the false positive control. Under the misspecified cell type compositions, STANCE-misspecified can still effectively control the type I error across different settings (Supplementary File II Figure S2), showcasing the robustness of STANCE in type I error control.

Like other ctSVG detection methods, STANCE's testing power is inherently linked to the precision of cell type deconvolution results, as it integrates both spatial information and cell type composition to detect ctSVGs. While STANCE demonstrates robustness against miss-deconvolved cell type composition for controlling the type I error, its testing power can be impacted when the deconvolution estimates are significantly misspecified. This equivalently applies to other ctSVG detection methods.

It is important to note that quantifying and evaluating the correctness of deconvolution results remains a complex challenge. This difficulty arises because there are no universally accepted benchmarks to assess the accuracy of cell type deconvolution in spatial transcriptomics data. Furthermore, while deconvolution is a critical preprocessing step for STANCE, the methodological details and potential limitations of deconvolution approaches are not the primary focus of STANCE. Instead, STANCE is designed to work with deconvolved data as input, assuming reasonable accuracy in the estimated cell type compositions, like other ctSVG detection methods do.

By maintaining this separation, STANCE ensures that its core methodology remains adaptable to various deconvolution techniques, while acknowledging that the quality of input data can influence its overall performance in ctSVG detection.

Similarly, three deconvolution methods (RCTD, CARD, STdeconvolve) are applied in three case studies, making it kind of cherry-picking in the process.

Thanks for the comments! The datasets used in the three case studies are publicly available and have been widely analyzed in prior research. We adhered to the deconvolution procedures established in the corresponding literature studies rather than selecting deconvolution techniques independently. This approach ensures consistency with existing methodologies and allows for a fair assessment of STANCE in various scenarios. For instance, in the case study involving mouse olfactory bulb data, we followed the workflow outlined in the original publication of STdeconvolve. By employing this approach, we aimed to demonstrate that STANCE is compatible with reference-free deconvolution techniques like

STdeconvolve and performs effectively when integrated with such methods. This highlights the versatility of STANCE in accommodating different deconvolution frameworks while maintaining robust and accurate results. In real applications, users can choose the specific deconvolution method they prefer.

8. In domain identification, incorporating SVGs using only spatialPCA with results at ARI 0.39, 0.42, 0.45 may not sufficiently claim they significantly improve domain detection.

Thank you for the comment! We added 4 other domain detection methods including both traditional machine learning and deep learning-based domain detection algorithms, such as `seuratPCA`, `BayesSpace`, `Stlearn` and `SpaceFlow`, besides `SpatialPCA`. The conclusion is consistent with `SpatialPCA`'s result, that `utSVGs` (combination of `ctSVGs` and general `SVGs`) work the best for domain detection purposes, and both `utSVGs` and `ctSVGs` outperform `SPARK SVGs` (see the figure below).

Additionally, we revised the wording related to domain detection analysis in the "Human Breast Cancer Data Analysis" subsection of the "Results" section to improve clarity and detail. Furthermore, we introduced a new subsection titled "Domain detection analysis" to elaborate on the technical aspects of this analysis. To complement this, we added Figure S11, which presents the results of the domain detection analysis, included in Supplementary File II.

9. What is the difference in methodology between STANCE and Celina (https://lulushang.org/Celina_Tutorial/index.html)? Even though Celina has not been officially published, it could be interesting to see how they differ.

Thank you for your insightful comment and suggestion! Since the manuscript for Celina has not yet been made available, our comparison between STANCE and Celina is based solely on the information provided on the Celina Tutorial website (https://lulushang.org/Celina_Tutorial/index.html).

Both STANCE and Celina employ a kernel matrix to capture spatial correlations and utilize linear mixed-effects models to investigate the relationship between spatial gene expression and spatial information. However, there are notable differences in their approaches:

1. Model structure

Celina focuses on detecting ctSVGs by incorporating a single cell-type-specific spatial random effect and a cell-type-specific non-spatial effect into its model. This streamlined structure enables a one-step testing procedure to identify ctSVGs for a specific cell type.

STANCE, on the other hand, incorporates K cell-type-specific random effects, allowing it to model all potential spatial relationships simultaneously. This comprehensive approach supports a broader analysis, including both ctSVGs and regular SVGs.

2. Detection capabilities

Celina is designed specifically for detecting ctSVGs, and its testing procedure outputs a ctSVG list for one cell type at a time.

In contrast, STANCE employs a two-step testing procedure. The first stage provides a unified list containing a list of utSVGs (could be SVGs or ctSVGs), while the second stage further distinguishes between these two categories, ctSVGs and SVGs.

3. Variance analysis and visualization

By modeling the cell type-specific variance components simultaneously in one model, STANCE offers additional functionality by estimating the relative proportions of variance explained by cell-type-specific effects and error variance. These proportions can be visualized through stacked bar plots, providing valuable insights into the contribution of different cell types. Celina does not provide this feature.

4. Computational efficiency and multiplicity issue

Suppose there are L cell types and total p genes. Celina requires separately testing cell type effect L times for each gene and ends up with $L \cdot p$ number of tests. This could be time-consuming (note that we did not conduct a comprehensive evaluation of their code). In addition, such a strategy can pose challenges for multiple testing control given that the $L \cdot p$ tests are not independent. Specifically, for each gene, the L tests are certainly not independent (the same gene is tested L times). For the p genes, they are not independent as well, given the complicated regulatory relationship among them. This makes the multiple testing corrections a challenging task.

For STANCE, the first-stage overall test, which uses a score statistic on a reduced model, is computationally efficient. The second-stage individual tests in STANCE may require additional computational time compared to Celina's framework; but given the reduced number of tests in the second stage, the computational burden remains

manageable. Furthermore, multiplicity adjustments are only needed for the p gene tests, as the first-stage test is a simultaneous test, thereby alleviating the multiple testing burden.

In summary, while Celina's streamlined model is advantageous for efficiently detecting ctSVGs, its scope is limited to one cell-type-specific analysis at a time. On the other hand, STANCE supports a unified analysis of both SVGs and ctSVGs, offering additional interpretability and flexibility.

10. How about the computational time and memory usage of the STANCE? It may be helpful to test its capacity in cellular-level sequencing data from 10X Xenium/Visium HD, et al. or directly state the analysis can be finished in seconds (minutes) on a desktop.

Thank you for your comment! The STANCE first-stage overall test, which employs a score statistic on a reduced model, is designed to be computationally efficient. However, the second-stage individual tests, which analyze each cell type separately, may require additional computational resources. In addition, the computational time scales approximately cubically with the number of spots due to the use of the kernel matrix, which may impose computational challenges for any kernel-based methods such as STANCE and SPARK.

For the MOB replicate #8 dataset comprising 7,365 genes, 260 spots, and 12 cell types (post-quality control and cell type deconvolution), the computational performance of various methods was evaluated on a desktop equipped with an Intel Core i7-13650HX CPU and 24 GB of RAM. The memory was calculated using the ```bench_memory``` function in the R `bench` package, which computes the cumulative memory usage. The results are summarized as follows:

SPARK-G: Completed in approximately 100 seconds (1.65 minutes), with a total memory usage of 3.63 GB.

SPARK-X: Completed in approximately 9.84 seconds, with a total memory usage of 170 MB.

STANCE stage 1 overall test: Completed in approximately 31.58 seconds, which includes 14.27 seconds for creating the STANCE object, normalization, constructing the kernel matrix, and computing covariance matrices for all variance components. The total memory usage was 448 MB.

STANCE stage 2 individual tests: Required approximately 985 seconds (16.4 minutes) to analyze all 828 utSVGs detected by the STANCE stage 1 test across all 12 deconvolved cell types. The total memory usage was 95.6 GB.

spVC: Required approximately 1.13 hours to complete the entire process, excluding the time needed for pre-selecting boundary points and creating triangulations based on the boundary. The total memory usage was 117 GB.

CTSV: Demonstrated intensive computational demands when analyzing all 7,365 genes. For comparison, it required approximately 5.16 minutes to process a subset of 10 genes. Taking the average time as 0.516 min/gene, it requires about 63 hours. The total memory usage for CTSV could not be measured using the ```bench_memory```.

We have included the above information in Section 2.4 "Computational time and memory usage" in Supplementary File II.

Minor comments:

1. In Figure 5B right, the legend says it is on fold change, but it looks like it is still at FDR like in Figure 5B left.

Thank you for pointing out this mistake. We moved the figure displaying power vs fold change to Supplementary File II and forgot to remove the corresponding caption. The error has been fixed.

2. The existing method 'CSIDE' should be 'C-SIDE'.

Thank you for your comment. We have updated the manuscript following your comment.

3. The manuscript uses the terminology 'utSVG' as a mixture of SVGs and ctSVGs. This term should be defined at the first time it occurs at the end of the introduction.

Thank you for your comment. Following your suggestion, we have updated the associated descriptions for STANCE and utSVGs as follows: "In the first stage, STANCE performs an overall test to identify the presence of SVGs and ctSVGs, classifying genes that pass this test as unified-type SVGs (utSVGs), which encompass both SVGs and ctSVGs. These utSVGs are then subjected to a second stage analysis for ctSVG detection."

4. In Section 4.3.1, the author mentioned "From the total N cells, we randomly selected three without replacement to serve as the centers for domains D2 and D3". Did the authors mean two points?

Thank you for your careful reading. Yes, it is a typo and has been fixed in the manuscript.

Reviewer #3 (Remarks on code availability):

I indeed install and run parts of the Rotation analysis on my windows machine but failed with SPARK on my Mac M1. The tutorial provided is kind of sufficient to me.

Suggestions: It may be better to provide required () function for installation, for some of the packages, such as SpatialExperiment and spVC, are not easy to install.

Thank you for your comment! Following your suggestion, we have updated the "Import Packages" section in Supplementary File I to include detailed instructions on how to install all the necessary dependencies.

Reviewer #4 (Remarks to the Author):

Thank you for your contribution to the review process!

Reviewer #1 (Remarks to the Author):

The authors have successfully addressed all my comments.

Thank you for your positive comment.

Reviewer #1 (Remarks on code availability):

The authors have updated the tutorials and functions according to the suggestions. The software should be suitable for public use.

Thank you for your positive comment.

Reviewer #2 (Remarks to the Author):

The authors have addressed all my questions.

Thank you for your positive comment.

Reviewer #2 (Remarks on code availability):

Software is user-ready.

Thank you for your positive comment.

Reviewer #3 (Remarks to the Author):

The authors significantly improved the quality of the manuscript, and they have already answered most of my questions. I only have one minor issue for my major review 2, so it would be better to show the results somewhere to demonstrate the invariance to the rotation when applying RCTD steps.

Thank you for your comment. RCTD employs a Poisson mixed-effects model to identify mixtures of cell types. Suppose there are n observed spots (pixels) across a tissue section. For a specific gene, the hierarchical model is given as:

$$y_i | \lambda_i \sim \text{Poisson}(N_i \lambda_i),$$
$$\log(\lambda_i) = \alpha_i + \log\left(\sum_{k=1}^K \beta_{i,k} \mu_k\right) + \gamma + \varepsilon_i,$$

where N_i is the total transcript count in spot i ; K is the number cell types; α_i is the fixed pixel-specific effect; μ_k is the mean gene expression for cell type k ; $\beta_{i,k}$ is the proportion of cell type k in spot i ; γ is the gene-specific platform random effect and ε_i is a random effect to account for other sources of variation.

Notably, this model does not incorporate spatial coordinate information. Consequently, changes in spatial coordinates, such as rotations, do not influence the results of RCTD's cell type deconvolution.

To demonstrate the rotational invariance of RCTD, we conducted the following validation (see also in the supplementary file):

We can verify that the estimated cell type compositions by RCTD remain invariant to rotation by the following step for validation.

```
# Check whether the estimated cell type compositions matrices from rotated patterns  
# are the same as the one from the original pattern.
```

```
all(myRCTD.original@results$weights == myRCTD.rotated.30@results$weights)
```

```
## [1] TRUE
```

```
all(myRCTD.original@results$weights == myRCTD.rotated.60@results$weights)
```

```
## [1] TRUE
```

```
all(myRCTD.original@results$weights == myRCTD.rotated.90@results$weights)
```

```
## [1] TRUE
```

The estimated cell type composition matrices produced by RCTD under 30°, 60°, and 90° rotations of the spatial pattern were compared to those obtained under the original pattern. We observed that the matrices were element-wise identical, with all entries matching exactly, confirming that RCTD's results are unaffected by spatial rotations. This result is provided in the supplementary file.

Reviewer #3 (Remarks on code availability):

The authors updated the codes according to my former recommendations

Thank you for your positive comment.

Reviewer #4 (Remarks to the Author):

Thank you for your contribution to the review process.